# *Ustilago maydis* disrupts carbohydrate signaling networks to induce hypertrophy in host cells

Yoon Joo Lee [1,5], Dong Zhang [1,5], Sara Christina Stolze [2], Georgios Saridis[1], Malaika K. Ebert [1,4], Hirofumi Nakagami [2,3] & Gunther Doehlemann [1]✉

*Ustilago maydis* infection in maize causes hypertrophic leaf tumors; however, the underlying mechanisms driving this excessive cell growth are unknown. In this study, we identify Hap1 (hypertrophy-associated protein 1) as an effector and virulence factor that regulates mesophyll cell hypertrophy. Using CRISPR-Cas9 mutagenesis, we demonstrate that Hap1 contributes to endoreduplication and starch accumulation in infected tissues. Transcriptomics revealed Hap1-dependent upregulation of starch biosynthesis and cell cycle genes, as well as suppression of plant defense. This links Hap1 to metabolic and cell cycle reprogramming, and immune suppression. To identify the target of Hap1 that drives metabolic reprogramming, we investigated its interaction with ZmSnRK1α in maize. We found that Hap1 interferes with the phosphorylation of SnRK1 substrates and that two Hap1-interacting effectors, Hip1 and Hip2, enhance its protein stability. We conclude that Hap1 contributes to the reprogramming of maize metabolism and cell cycle, as well as mesophyll cell hypertrophy, by modulating the SnRK1 signaling pathway to regulate starch biosynthesis and host defense responses.

Smut fungi are one of the largest groups of biotrophic plant pathogens, infecting a wide range of cereal plants[1]. Most smuts spread disease systemically through the vascular system, causing symptoms that are confined to the inflorescences[2,3]. In contrast, *Ustilago maydis* forms local tumors in all aerial parts of maize[4]. To infect maize organs, which vary substantially in their development and physiology, *U. maydis* secretes a cocktail of effectors in a spatiotemporal and organ-specific manner, promoting host colonization and suppressing plant immune responses[5,6].

In vivo visualization of *U. maydis*-induced tumors showed that newly divided bundle sheath cells transform into hyperplastic tumor (HPT) cells, while mesophyll cells enlarge and become hypertrophic tumor (HTT) cells due to endoreduplication, a cellular mechanism characterized by DNA replication without cell division, resulting in an increased nuclear DNA content[7,8] Targeted transcriptomic analysis using laser-captured microdissection identified specific effector gene expressions associated with HPT and HTT formation, highlighting the cell-type specific role of effectors in tumor formation[7]. For example, See1 (Seedling efficient effector 1) interacts with ZmSGT1, a cell cycle transition regulator, to reactivate DNA synthesis essential for cell division in HPT cells[7]. More recently, the effector Sts2 (Small Tumor on Seedlings 2), which promotes HPT induction, was found to interact with ZmNECAP1, a plant transcriptional activator, thereby activating leaf development regulators to potentiate tumor formation[8]. However, effectors involved in HTT formation and their underlying molecular mechanisms remain uncharacterized.

*U. maydis* relies on living tissues for survival, invading the intercellular spaces by breaking down the loosened cell wall to extract

[1]Institute for Plant Sciences and Cluster of Excellence on Plant Sciences (CEPLAS), University of Cologne, Cologne, Germany. [2]Protein Mass Spectrometry, Max-Planck Institute for Plant Breeding Research, Carl-von-Linné Weg 10, Cologne, Germany. [3]Basic Immune System of Plants, Max Planck Institute for Plant Breeding Research, Cologne, Germany. [4]Present address: Department of Plant Pathology, North Dakota State University, NDSU Department 7660, Fargo, ND, USA. [5]These authors contributed equally: Yoon Joo Lee, Dong Zhang. ✉e-mail: g.doehlemann@uni-koeln.de

nutrients from the host[9]. Upon infection, it reprograms plant metabolic processes, redirecting starch toward developing tumors, thereby transforming them into strong nutrient sinks[10–13]. Notably, starch granules begin to excessively accumulate as early as 2 dpi, particularly in mesophyll HTT cells, which are atypical sites for starch storage in C4 plants[7,14]. Moreover, several transcriptomic analyses have also shown that the infection leads to extensive transcriptional changes in maize, including up-regulation of glycolysis, the tricarboxylic acid (TCA) cycle, and defense-related genes, as well as down-regulation of photosynthetic genes. The regulation of plant metabolism is crucial, especially when challenged by stress. Sucrose-nonfermenting1 (SNF1)-related protein kinase 1 (SnRK1) functions as a central energy regulator, consisting of a catalytic α subunit and regulatory β and γ subunits[15]. SnRK1 regulates growth and developmental transitions under varying stresses and energy conditions by phosphorylating key transcription factors and metabolic enzymes[16–19]. Its kinase activity is activated by the phosphorylation of a conserved threonine residue in the T-loop of the catalytic domain, mediated by upstream kinases (SnAK1/2) or via autophosphorylation[15,20,21]. SnRK1 regulates a broad range of signaling and metabolic pathways by promoting catabolic processes to mobilize storage compounds, while repressing anabolic processes. In addition, SnRK1 modulates plant immunity against diverse plant pathogens and is targeted by pathogenic effector proteins[22]. For example, in pepper, SnRK1 suppressed the AvrBs1-specific hypersensitive response mediated by the effector protein AvrBsT in *X. campestris* pv. *vesicatoria*[23]. In wheat, TaSnRK1 was shown to interact with *Fusarium graminearum* orphan secreted protein OSP24 and to be important for Fusarium head blight resistance[24]. Overexpression of SnRK1A in rice enhanced resistance to broad-spectrum hemibiotrophic and necrotrophic pathogens[25]. *Plasmodiophora brassicae*-specific effector PBZF1 inhibited SnRK1.1-mediated resistance in *Arabidopsis thaliana* against clubroot disease[26].

In this study, we identified the effector Hap1 as a virulence factor in *U. maydis* that regulates hypertrophic tumor formation by promoting endoreduplication and starch accumulation. We show that Hap1 interacts with maize SnRK1α, and that two Hap1-interacting effectors, Hip1 and Hip2, enhance this interaction. Phosphoproteomics data and SnRK1 activity assay indicate that Hap1 is required for specific phosphorylation events on SnRK1 subunits and phosphorylation of SnRK1 downstream substrates. Overall, these results highlight an effector-mediated mechanism for host metabolic reprogramming during pathogen-induced tumor development.

## Results

### Characterization of candidate effectors associated with hypertrophic tumor formation in Ustilago maydis

To study the molecular functions of HTT effectors, we selected ten HTT-specific candidate genes that were previously identified as being only expressed in HTT cells and not in HPT cells, suggesting a specialized role in the development of hypertrophic tumors[7] (Table 1). Among these candidates, *Cda7* and *Sts2* were excluded due to their established functional roles as a chitin deacetylase and a transcriptional activator, respectively[8,27]. *UMAG_00753* and *UMAG_10642* were excluded for lacking a conventional N-terminal signal peptide.

To determine whether the remaining six effector candidates exhibit virulence (Supplementary Data 1), open reading frame shift knockout mutants for each gene were generated using CRISPR-Cas9 mutagenesis in the solopathogenic *U. maydis* strain SG200[28]. The resulting mutants were tested in maize seedling infections: while four of the mutants did not exhibit significant reduced virulence (Supplementary Fig. 1), the CR-UMAG _02473 and CR-UMAG_00793 mutants displayed a significant reduction in tumor formation (Supplementary Fig. 1 and Fig. 1a). The genetic complementation of CR-UMAG_02473 and CR-UMAG_00793 mutants with the native gene, using its own promoter, restored virulence. This confirmed the gene's role in virulence in this host background

**Table 1 | Overview of HTT-related effector candidates tested in this study**

| Effector | Gene ID | SP | TPMmax | Putative activity | Reference |
|---|---|---|---|---|---|
| **Effector candidates with HTT specific expression** | | | | | |
| Cda7 | UMAG_02381 | Y | 12 dpi | Chitin deacetylase | Rizzi et al.[27] |
| – | UMAG_05222 | Y | 1 dpi | – | – |
| – | UMAG_00753 | N | 1 dpi | – | – |
| – | UMAG_00793 | Y | 2 dpi | – | – |
| – | UMAG_02473 | Y | 2 dpi | – | – |
| Sts2 | UMAG_05318 | Y | 4 dpi | Transcription activator | Zuo et al.[8] |
| – | UMAG_10642 | N | 0.5 dpi | – | – |
| – | UMAG_03650 | Y | 6 dpi | – | – |
| – | UMAG_12119 | Y | 1 dpi | – | – |
| – | UMAG_11484 | Y | 6 dpi | – | – |

This table presents an overview of *Ustilago maydis* effector candidates related to hypertrophic tumor cells (HTT) evaluated in this study. *SP* signal peptide. TPMmax, the highest Transcripts Per Million, reflecting the peak expression levels of these effectors at various plant-associated time points throughout different growth stages[32].

(Supplementary Fig. 1 and Fig. 1a). Between the two, *UMAG_02473* exhibited a much higher expression level in HTT than *UMAG_00793*[7]. Therefore, we selected *UMAG_02473* for further functional studies.

### Hap1 regulates hypertrophic tumor formation

Endoreduplication is a hallmark of hypertrophic cell formation, and starch is known to accumulate strongly, particularly in hypertrophic cells[7,14]. We therefore assessed whether the effector UMAG_02473 promotes HTT formation by examining both endoreduplication and starch accumulation in maize seedlings infected with different *U. maydis* strains: mock, SG200, CR-UMAG_02473, and CR-sts2. *U. maydis* strain CR-sts2, in which the transcription activator effector Sts2 that promotes de novo cell division in HPT is disrupted, was used as a positive control. Endoreduplication was evaluated by staining leaf sections with propidium iodide (PI) and measuring nuclear size in mesophyll cells. Interestingly, tissues infected with CR-UMAG_02473 displayed nuclear sizes similar to mock, but approximately two-fold smaller than those infected with SG200 and CR-sts2 (Fig. 1b), suggesting that UMAG_02473 promotes endoreduplication in mesophyll cells. To examine how UMAG_02473 affects starch distribution, tissue sections were stained with Lugol's iodine solution (IKI). In mock-infected tissues, starch was confined to the bundle sheath, whereas SG200-infected tissues showed dispersed and increased starch accumulation in mesophyll cells (Fig. 1c–f). This pattern is consistent with previous studies showing that the induction of hypertrophy in maize mesophyll cells is accompanied with elevated starch accumulation[7], suggesting that the infection alters carbohydrate metabolism and affects chloroplast dimorphism. Notably, CR-UMAG_02473-infected tissues displayed significantly reduced starch accumulation in mesophyll compared to SG200, whereas CR-sts2 showed no significant difference from SG200 (Fig. 1d–g), indicating that reduced starch accumulation is a specific characteristic of CR-UMAG_02473. These results demonstrate that UMAG_02473 promotes both endoreduplication and starch accumulation in mesophyll cells and thereby regulates hypertrophic tumor formation. We therefore named UMAG_02473 as Hap1 (Hypertrophy-associated protein 1).

### Hap1 mediates transcriptional control of carbohydrate metabolism in maize during U. maydis infection

To investigate the host transcriptional response specifically triggered by Hap1 during infection, particularly related with endoreduplication and starch accumulation, RNA sequencing was performed on maize

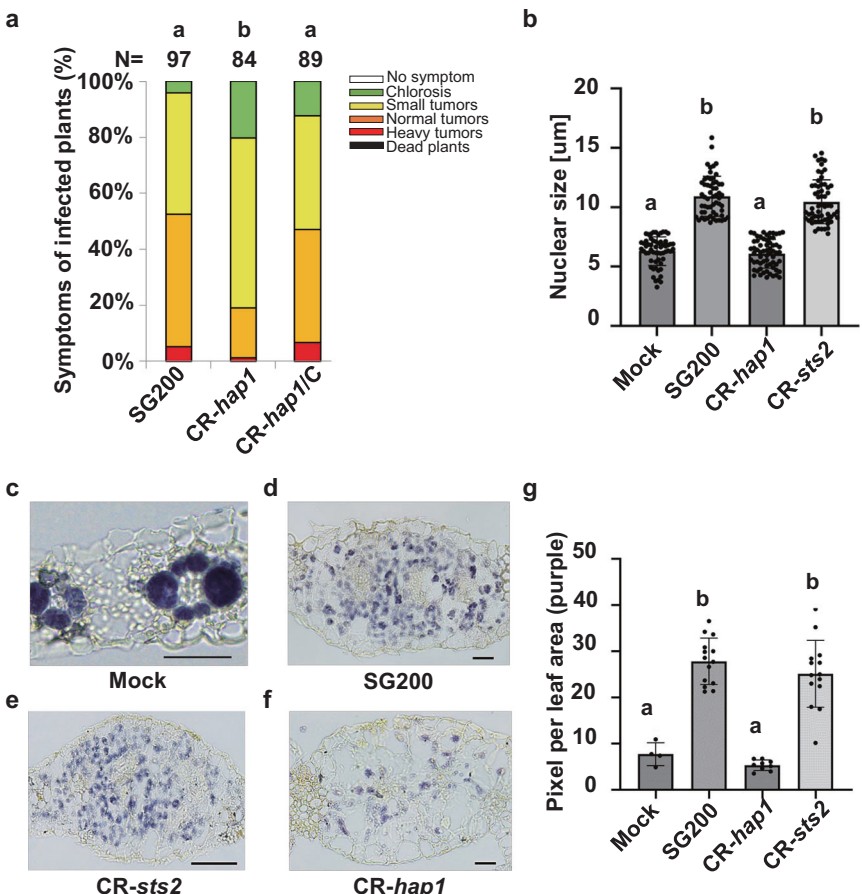

**Fig. 1 | Hap1 contributes to *U. maydis* virulence and maize hypertrophic tumor formation.** The Hap1 protein is encoded by gene *UMAG_02473*. **a** Disease symptoms in maize infected with *U. maydis* frameshift knockout mutant CR-*hap1* and complemented strain CR-*hap1*/C are compared to the wild-type strain SG200 at 12 dpi. The disease index is presented as the mean from three biological replicates. Statistical significance was determined using one-way ANOVA with Tukey's HSD test ($P < 0.05$). Different letters indicate statistically significant differences among treatments. **b** Quantification of histological cross-sections of leaf tissues infected with mock or *U. maydis* (SG200, CR-*hap1*, CR-*sts2*) stained with propidium iodide at 6 dpi. Quantification was performed from three independent biological replicates, each with > 55 nuclei per sample. **c**–**e** Histological cross-sections of leaf tissues infected with mock or *U. maydis* (SG200, CR-hap1, CR-sts2) stained with Lugol iodide (IKI) at 6 dpi: (**c**) Mock, (**d**) SG200, (**e**) CR-hap1, (**f**) CR-sts2. Representative image of SG200, CR-hap1, and CR-sts2 from three independent biological replicates and mock from two independent biological replicates. Scale bars = 50 μm. **g** Quantification of starch staining from Fig.1c–f. Statistical analysis for panels (**d** and **j**) were performed used one-way ANOVA with Tukey's HSD test ($P < 0.05$). Different letters indicate statistically significant differences among treatments.

seedlings infected with mock, SG200, or the mutant CR-*hap1* at 3 dpi, using three independent biological replicates. Compared to mock-infected plants, SG200 infection (S vs. M) triggered 12,492 differentially expressed genes (DEGs) (7676 up-regulated and 4816 down-regulated), while CR-*hap1* infection (CR-H vs. M) resulted in 10,603 DEGs (6,795 up-regulated and 3,808 down-regulated) (Supplementary Fig. 2 and Supplementary Data 1, 2), indicating a slightly attenuated host response to the mutant strain. Of these, 5,974 up-regulated and 3,072 down-regulated DEGs were common to both comparisons (Supplementary Fig. 2b, c), representing a core infection response. To pinpoint Hap1-specific effects, a comparison between CR-*hap1* and SG200 treatment (CR-H vs. S) identified 2339 genes, of which 908 were up-regulated and 1431 were down-regulated (Fig. 2a and Supplementary Data 3).

Gene ontology (GO) enrichment analysis revealed that oxidative stress-related responses were significantly up-regulated in the S vs. M and CR-H vs. M, with a stronger enrichment observed in the latter comparison (Supplementary Data 4, 5). The term "response to oxidative stress" (GO:0006979) was similarly enriched in both comparisons, whereas "peroxidase activity" (GO:0004601) and "ethylene-activated signaling pathway" (GO:0009873) were more enriched in CR-H vs. M (Supplementary Figs. 3, 4). This suggests that Hap1 partially suppresses ROS-associated and ethylene-mediated defense responses. Ribosome-related

biological processes, such as "small subunit assembly" (GO:0000028) and "nucleolus" (GO:0005730) were significantly down-regulated in CR-H vs. S (Supplementary Fig. 5), indicating reduced ribosome assembly activity in the absence of Hap1. In addition, "cytosolic large ribosomal subunit" (GO:0022625) and "translational elongation" (GO:0006414) were also down-regulated in CR-H vs. S (Supplementary Fig. 5 and Supplementary Data 6), suggesting attenuation of ribosome biogenesis and protein synthesis in Hap1 mutant infection.

SG200 infection led to strong up-regulation of 104 cell cycle-related genes, particularly those governing the G1/S phase transition and DNA replication. These including WEE1 kinase, siamese-related proteins SMR1 and SMR3, multiple D-type cyclins, and components of the pre-replication complex (MCM3-7, ORC5-6, CDC6, CDT1) as well as genes encoding replication proteins such as RPA1-3, DNA polymerase alpha subunit POLA2, and E2F-DP transcription factors essential for S phase onset were up-regulated (Fig. 2b and Supplementary Data 7). In the CR-H vs. S comparison, GO terms like "DNA replication initiation" (GO:0006270), "Nucleosome assembly" (GO:0006334), and "nucleosome" (GO:0000786) were down-regulated (Supplementary Data 6), indicating Hap1 drives S-phase progression.

In addition, SG200 infection increased expression of starch biosynthesis genes, including AGPase3, SBE I, and Ae1 (Fig. 2c and Supplementary Data 7). Correspondingly, the GO term "starch metabolic

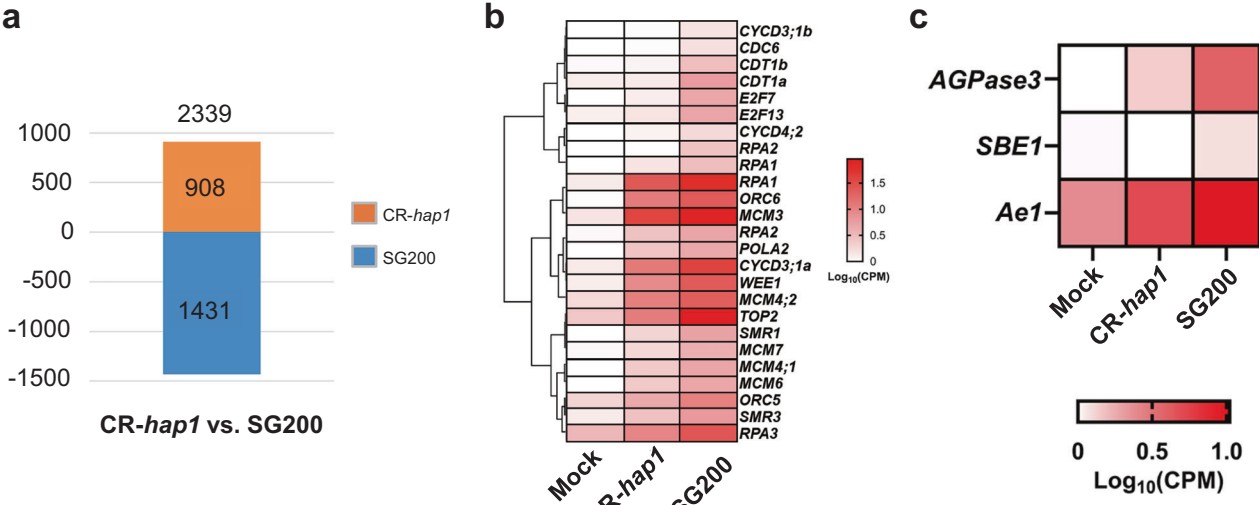

**Fig. 2 | Maize genes specifically influenced by the *U. maydis* Hap1 effector during Infection. a** Pairwise bar graph depicting the number of up- and down-regulated DEGs in CR-*hap1* vs. SG200. **b** Heatmap of DEGs related to cell cycle processes, specifically involved in G-S1 phase transition, including WEE1 kinase, SMR1 (siamese-related1), SMR3 (siamese-related3), three D-type cyclins, and pre-replication complex (pre-RC) components, mini-chromosome maintenance 3–7 (MCM3-7), Origin recognition complex 5-6 (ORC5-6), cell division cycle 6 (CDC6), Cdc10-dependent transcript 1 (CDT1), as well as replication protein A 1-3 (RPA1-3), DNA polymerase alpha subunit 2 (POLA2), and two E2F-DP coding genes required for the onset of the S phase. DEGs = Log2FC > 1 or < − 1 and FDR < 0.05. **c** Heatmap of DEGs related to key enzymes ADP-glucose pyrophosphorylase 3 (AGPase 3), starch-branching enzyme I (SBE I), and amylose extender 1 (Ae1)) in starch bio-synthesis in SG200-infected samples. DEGs = Log2FC > 1 or <− 1 and FDR < 0.05.

process" (GO:0005982) was down-regulated in CR-H vs. M but not in S vs. M (Supplementary Figs. 3, 4 and Supplementary Data 4, 5), highlighting the role of Hap1 in promoting starch accumulation. In contrast, the "carbohydrate derivative metabolic process" (GO:1901135) was up-regulated in CR-H vs. S (Supplementary Data 6), suggesting a shift toward alternative sugar metabolism in the absence of Hap1.

## Hap1 interacts with SnRK1α

To identify host targets, we firstly tested if the effector is delivered into host cells. *U. maydis* strain CR-*hap1* was transformed for stable expression of the respective 2xHA-tagged effector under control of the *pit2* promoter, which confers strong expression in infectious hypha[29]. The resulting recombinant *U. maydis* strain (SG200-CR-*hap1*-p*Pit2*::Hap1-2xHA) was used for infection of maize leaves and subsequent protein isolation at 3 dpi. After subcellular fractionation, Hap1−HA was detected in cytoplasmic and nuclear fractions together with maize UDP-Glucose-Pyrophosphorylase (UGPase) and histone H3, respectively (Fig. 3a), indicating that Hap1 accumulates inside plant cells. In the nuclear fraction, Hap1-2HA resolved predominantly as a slightly higher-molecular-weight band along with a faint band at the expected size (Fig. 3a), suggesting post-translationally modified forms of the effector in the nucleus.

We next proceeded to identify the maize proteins targeted by Hap1 that may mediate these transcriptional changes. We performed immunoprecipitation (IP) followed by mass spectrometry (MS) analysis. While the Hap1-expressing strains (SG200-CR-*hap1*-p*Pit2*::Hap1-2xHA) were used for target identification, a *U. maydis* strain expressing HA-tagged mCherry with a signal peptide (SG200-p*Pit2*::SP-mCherry-HA) was used as a control (Fig. 3b). The identified spectra from the IP-MS/MS analyses were mapped to the maize proteome (Fig. 3b, Mass-spectrometry results in Supplementary Data 8). Top 150 proteins detected in Hap1-expressing samples were analyzed with GO enrichment using PLAZA 5.0 and grouped into higher hierarchical terms by REVIGO, revealing that Hap1 targets maize proteins involved in protein phosphorylation and serine-threonine kinase activity, particularly isoforms of the SnRK1α subunit SnRK1.1α (Zm00001eb013270) SnRK1.2α (Zm00001eb293240) SnRK1.3α (Zm00001eb094400) (Supplementary Fig. 6), a metabolic regulator known to regulate starch biosynthesis.

To confirm an interaction between Hap1 and ZmSnRK1α, a split-luciferase complementation assay was conducted. In this assay, Hap1^ΔSP-cLuc was transiently co-expressed with nLuc-ZmSnRK1α1, -ZmSnRK1α2, or -ZmSnRK1α3 in *N. benthamiana* using *Agrobacterium*-mediated transformation. Luminescence signals were detected exclusively when Hap1^ΔSP-cLuc was co-expressed with nLuc-ZmSnRK1α2, while no visible signal was observed with nLuc-ZmSnRK1α1 and nLuc-ZmSnRK1α3, empty-cLuc, or -nLuc controls (Fig. 3c, d). To further validate the interaction between Hap1 with ZmSnRK1α, co-immunoprecipitation (Co-IP) assays in *N. benthamiana* were conducted. *Agrobacterium* strain carrying Hap1^ΔSP-6xHA were co-infiltrated with ZmSnRK1α1-4xMyc, ZmSnRK1α2-4xMyc, or ZmSnRK1α3-4xMyc. As a negative control, GFP-4xMyc was co-infiltrated with Hap1^ΔSP-6xHA. Agrobacterium-infiltrated leaves of ZmSnRK1α1, ZmSnRK1α2, or ZmSnRK1α3 were co-immunoprecipitated by α-Myc immunoprecipitation of Hap1^ΔSP-6xHA, but not in GFP-4xMyc control (Fig. 3e). These results show that Hap1 interacts with catalytic subunits of ZmSnRK1.

## Hap1 interacts with U. maydis effectors Hip1 and Hip2

In addition to maize proteins, IP-MS also identified an enrichment of *U. maydis* effector proteins that potentially interact with Hap1 (Supplementary Data 9). Subsequent analyses using EffectorP_Fungi 3.0, SignalP 6.0, TPMmax, and InterproScan identified secreted cytoplasmic effectors with the highest expression at 2 dpi[30–33]. Among these effectors, we found that one effector, *UMAG_00793*, is specifically expressed in maize hypertrophic mesophyll tumor cells during *U. maydis* infection (Table 1) and contributes to virulence (Supplementary Fig. 1). Another effector, *UMAG_00792*, is a paralog of *UMAG_00793*, sharing 90% alignment with 28.12% protein sequence identity. The two genes are located on chromosome 1 and share an 848 bp intergenic promoter region, suggesting co-expression (Supplementary Fig. 7). To test potential redundancy between the two effectors, we also assessed the virulence of *UMAG_00792* (Supplementary Fig. 8) and generated and a strain (SG200ΔUMAG_00793::NP-*UMAG_00792*-2xHA), in which UMAG_00792 genetically complements the *UMAG_00793* knockout for a virulence test in maize seedlings

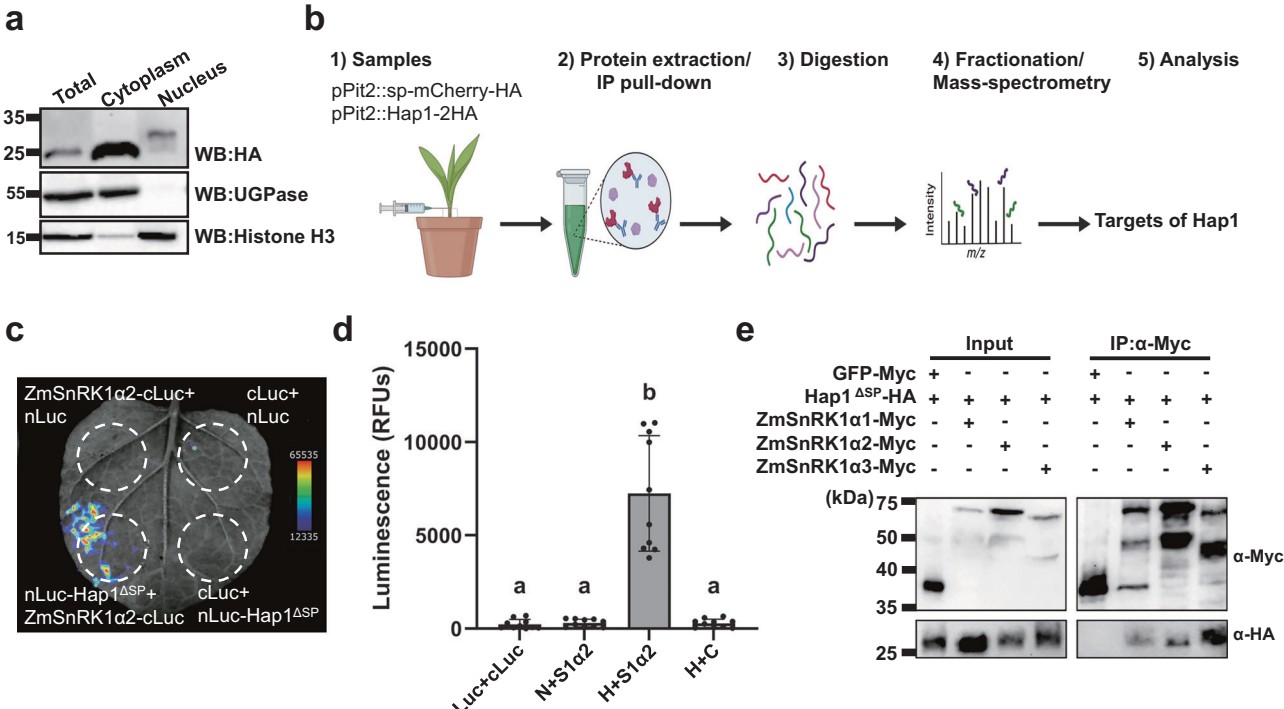

**Fig. 3 | Hap1 interacts with maize SnRK1α. a** Hap1 is translocated into host cells. Hap1-2 × HA detected by western blot in maize leaf cytoplasmic and nuclear fractions at 3 dpi following infection. UDP-Glucose-Pyrophosphorylase (UGPase) and histone H3 were used as the markers of cytoplasmic and nucleus fractions, respectively. The positions of the molecular weight ladder are shown on the left. **b** Overview of pull-down/mass spectrometry (MS) workflow for identifying effector targets. (1) Seven-day-old maize seedlings were infected with *U. maydis* SG200 strains expressing HA-tagged Hap effectors and collected at 3 dpi. (2) Total proteins were extracted and immunoprecipitated using HA magnetic beads. (3) Proteins bound to the beads were digested. (4) Digested peptides were fractionated and analyzed by MS. (5) Identified spectra were mapped against the maize genome to identify interacting effector proteins. Created in BioRender. Doehlemann, G. (2026) https://BioRender.com/1zmsye4. **c** Split-luciferase complementation assay in *N. benthamiana*. nLuc-Hap1^ΔSP or empty-nLuc were co-expressed with ZmSnRK1α2-cLuc, ZmSnRK1α3-cLuc, or empty-cLuc. Luminescence was detected using Bio-Rad ChemiDoc™ and pseudo-fluorescence applied for enhanced visualization. Representative image of ZmSnRK1α2 from three independent biological replicates and ZmSnRK1α3 from two biological independent replicates. **d** Quantification of luciferase activity, shown as relative luciferase units (RFUs), from the data presented in Fig. 4A. Error bars represent standard deviation; $n = 3$ biological replicates. Statistical analysis was performed using one-way ANOVA with Tukey's HSD test ($P < 0.05$). Different letters indicate statistically significant differences among treatments. N: nLuc; C: cLuc; H: Hap1; S1α2: ZmSnRK1α2. **e** Co-immunoprecipitation (Co-IP) assay in *N. benthamiana*. p2x35S-Hap1^ΔSP-6xHA was co-expressed with p2x35S- ZmSnRK1α1-4xMyc, p2x35S- ZmSnRK1α2-4xMyc, or p2x35S- ZmSnRK1α3-4xMyc and controls with p2x35S-Hap1^ΔSP-6xHA with p2x35S-GFP-4xMyc. Proteins pulled down with Myc magnetic beads were detected using anti-HA or anti-Myc antibodies. Expected protein sizes: Hap1^ΔSP-6xHA = 20.07 kDa; ZmSnRK1α1-4xMyc = 64.23 kDa; ZmSnRK1α2-4xMyc = 63.27 kDa; ZmSnRK1α3-4xMyc = 62.96 kDa; GFP-4xMyc = 31.8 kDa.

(Supplementary Fig. 9). Knockout of *UMAG_00792* resulted in significant reduction in virulence (Supplementary Fig. 8), however, strain SG200ΔUMAG_00793::NP-*UMAG_00792*-2xHA failed to restore this virulence defect, displaying a disease incidence similar to the UMAG_00793 knockout (Supplementary Fig. 9). These results indicate that *UMAG_00792* and *UMAG_00793* are not functionally redundant. Next, we asked whether Hap1 may physically interact with UMAG_00792 and/or UMAG_00793. To test the interaction between Hap1 and the other two effectors (UMAG_00793 and UMAG_00792), IP-MS was conducted for UMAG_00793 and UMAG_00792 using the same assay as for Hap1. UMAG_00792 pulldown enriched 4 proteins and captured Hap1 and UMAG_00793, while UMAG_00793 pulldown enriched 13 proteins and retrieved Hap1(Supplementary Data 10, 11), demonstrating that both UMAG_00792 and UMAG_00793 physically associate with Hap1.

To verify the interactions of Hap1 and the other two effectors, a split luciferase complementation assay was performed. In this assay, nLuc-Hap1^ΔSP, - UMAG_00792^ΔSP, or - UMAG_00793^ΔSP was transiently co-expressed with Hap1^ΔSP-, UMAG_00792^ΔSP-, or UMAG_00792^ΔSP-cLuc in *N. benthamiana* using *Agrobacterium*-mediated transformation. Empty-nLuc or -cLuc were used as a control. Luminescence signal was detected when Hap1 was expressed with UMAG_00792 or UMAG_00793, or when UMAG_00792 and UMAG_00793 were co-

expressed, whereas no signal was observed with any effector co-expressed with empty cLuc or nLuc controls (Fig. 4a–f). For another line of evidence, Co-IP were performed using α-Myc or α-GFP magnetic beads in *N. benthamiana*. *Agrobacterium* strain carrying Hap1^ΔSP-GFP or UMAG_00792^ΔSP-4xMyc was co-infiltrated with UMAG_00793^ΔSP-6xHA or Hap1^ΔSP-6xHA. As a negative control, eGFP or GFP-4xMyc was co-infiltrated with Hip2^ΔSP-6xHA or Hap1^ΔSP-6xHA. Hap1^ΔSP-GFP or UMAG_00792^ΔSP-4xMyc was co-immunoprecipitated with Hap1^ΔSP-6xHA or UMAG_00793^ΔSP-6xHA, whereas no HA-tagged proteins were pulled down when eGFP or GFP-4×Myc controls were used (Fig. 4g–i). Collectively, these results confirm that Hap1 physically interacts with UMAG_00792 and UMAG_00793 in vivo, and accordingly, we renamed UMAG_00792 and UMAG_00793 as Hip1 and Hip2 (Hap1 interacting proteins).

Given the overlapping, HTT-specific expression patterns of Hap1, Hip1, and Hip2 at 2 dpi (Table 1), as well as their interactions *in planta*, we hypothesized that these effectors might have additive or cooperative effects on virulence and starch accumulation. To test this hypothesis, we generated a triple frameshift knockout mutant CR-*hap1/hip1/hip2* using CRISPR-Cas9 mutagenesis. Interestingly, the virulence level of the resulting triple mutant was similar to that of CR-*hap1* (Supplementary Fig. 10), indicating that under our experimental conditions, Hip1 and Hip2 did not measurably affect virulence in the

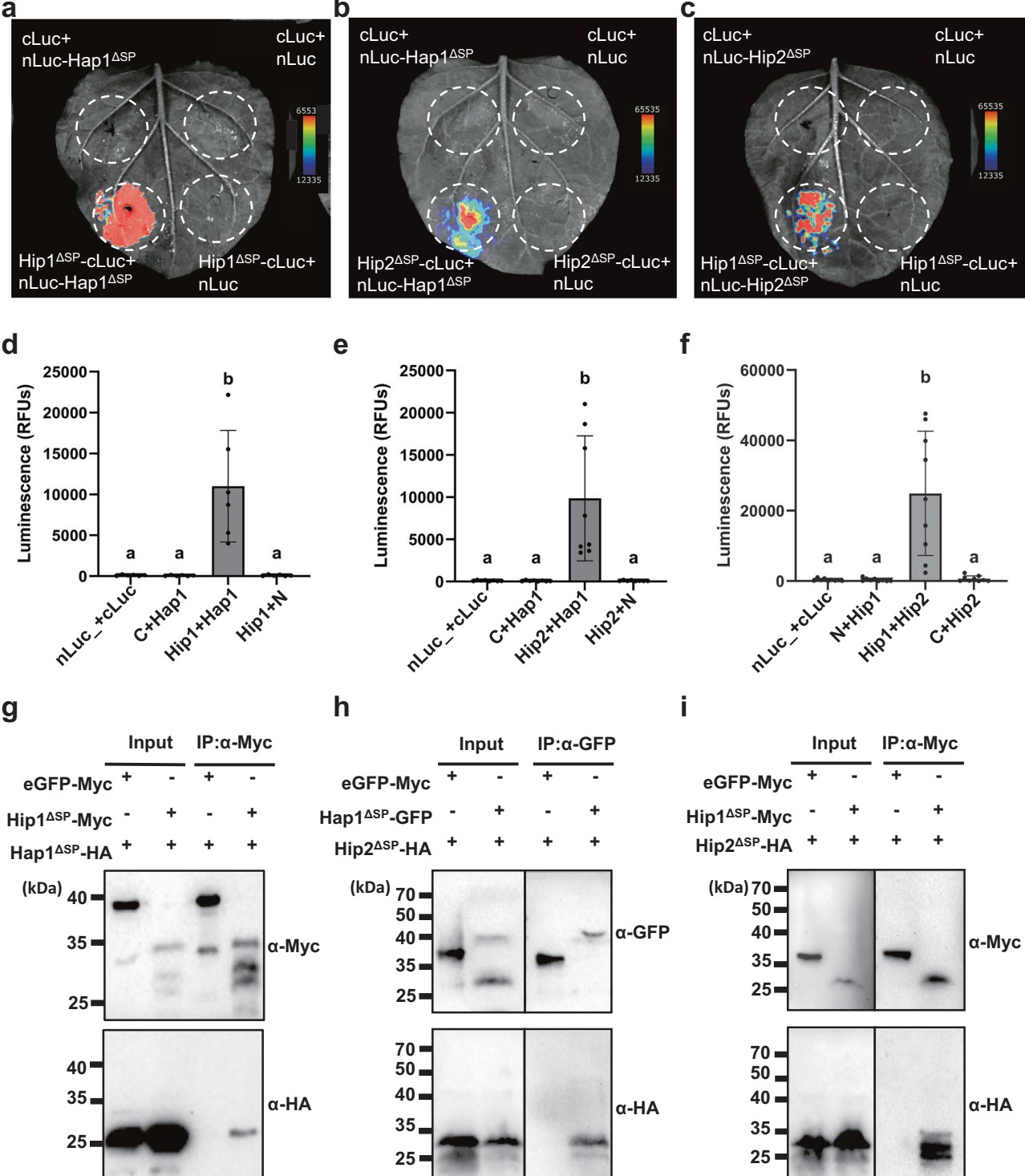

**Fig. 4 | Hap1 interacts with Hip1 and Hip2.** Hip1 and Hip2 are encoded by genes UMAG_00792 and UMAG_00793, respectively. **a**–**c** Split-luciferase complementation assay in *N. benthamiana* for confirmation of effector interaction. nLuc-Hap1$^{\Delta SP}$, nLuc-Hip1$^{\Delta SP}$, nLuc-Hip2$^{\Delta SP}$, or empty-nLuc were co-expressed with Hap1$^{\Delta SP}$-cLuc, Hip1$^{\Delta SP}$-cLuc, Hip2$^{\Delta SP}$-cLuc, or empty-cLuc. Luminescence was detected using Bio-Rad ChemiDoc™ and pseudo-fluorescence applied for enhanced visualization. A representative image from three independent replicates is shown.
**d**–**f** Quantification of luciferase activity, shown as relative luciferase units (RFU), from the data presented in Fig. 3C. Error bars represent standard deviation; $n = 3$ biological replicates. Statistical analysis was performed using one-way ANOVA with Tukey's HSD test ($P < 0.05$). Different letters indicate statistically significant differences among treatments. **g**–**i** Co-immunoprecipitation (Co-IP) assay in *N. benthamiana*. p2x35S-Hap1$^{\Delta SP}$-GFP or p2x35S-Hip1$^{\Delta SP}$-4xMyc were co-expressed with p2x35S-Hap1$^{\Delta SP}$-6xHA or p2x35S-Hip2$^{\Delta SP}$-6xHA, and controls with p2x35S-Hap1$^{\Delta SP}$-6xHA with p2x35S-GFP or p2x35S-GFP-4xMyc. Proteins pulled down with GFP or Myc magnetic beads were detected using anti-HA, anti-GFP, or anti-Myc antibodies. Expected protein sizes: Hap1$^{\Delta SP}$-GFP = 40.05 kDa; Hip1$^{\Delta SP}$-4xMyc = 23.11 kDa; Hap1$^{\Delta SP}$-6xHA = 20.07 kDa; Hip2$^{\Delta SP}$-6xHA = 23.22 kDa; GFP = 27 kDa; GFP-4xMyc = 31.8 kDa. N: nLuc; C: cLuc.

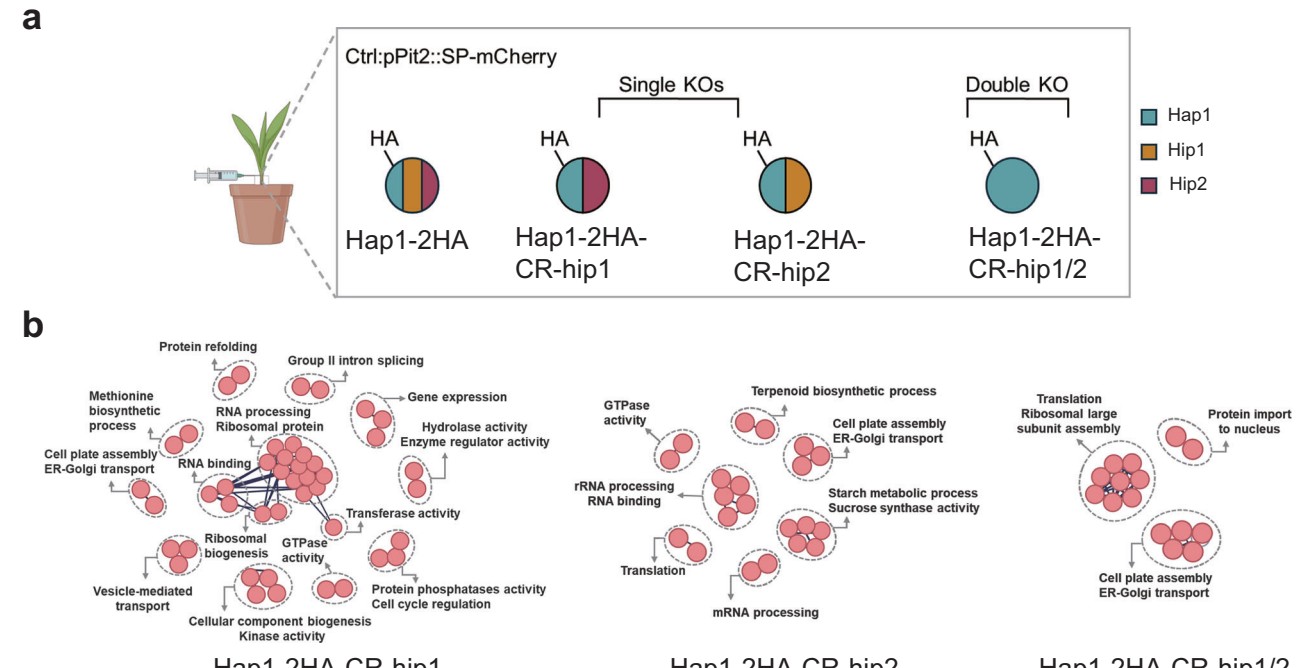

**Fig. 5 | Hip1 and Hip2 influence the Hap1 interactome. a** IP/MS was conducted for host target identification of Hap1-CR-*Hip1*, Hap1-CR-*Hip2*, and Hap1-CR-*Hip1/2*. 7-day-old maize seedlings were infected with *U. maydis* SG200 strains expressing SG200-CR-*Hap1*-p*Pit2*::Hap1-2xHA-CR-*Hip1*, SG200-CR-*Hap1*-p*Pit2*::Hap1-2xHA-CR-*Hip2*, SG200-CR-Hap1-p*Pit2*::Hap1-2xHA-CR-*Hip1-2*, or SG200-p*Pit2*::SP-mCherry-HA. Blue represents Hap1 protein, yellow represents Hip1 protein, and red represents Hip2 protein. The inoculation icon was created in BioRender. Doehlemann, G. (2026) https://BioRender.com/1zmsye4. **b** The PPI network was constructed using the STRING database, with nodes demonstrating individual proteins and edges representing predicted functional associations and interactions. Default settings with a high confidence threshold (0.7) were applied, and the network was clustered using K-means clustering. The connections shown are manually curated to highlight important interactions.

absence of Hap1. To assess whether Hip1 and Hip2 contribute to starch accumulation, we analyzed CR-hip1 and CR-*hap1/hip1/hip2*. Both the CR-*hap1* and CR-*hap1/hip1/hip2* showed significantly reduced starch accumulation in mesophyll cells compared to SG200 and single mutant of sts2 (Supplementary Fig. 11b–d and g). Although CR-h*ip1* also showed a reduction in starch levels compared to SG200, this difference was not significant when compared to the CR-*sts2*, which CR-*sts2* itself did not differ significantly from SG200 in starch accumulation (Supplementary Fig. 11a, b, e, f, and g). These findings suggest that the reduction in starch accumulation is a specific phenotype linked to the absence of Hap1 and not a general consequence of any effector deletions.

## Hip1 and Hip2 influence Hap1-ZmSnRK1α interaction and Hap1 interactome

Because Hap1 co-immunoprecipitates with Hip1 and Hip2 from infected maize tissue, and all three pairwise interactions (Hap1-Hip1, Hap1-Hip2, and Hip1-Hip2) were validated *in planta* in *N. benthamiana*, we hypothesized that Hip1 and Hip2 may mediate interactions between Hap1 and host proteins. To test this, we generated single frameshift knockout mutants for *hip1* and *hip2* (SG200-CR-*hap1*-p*Pit2*::Hap1-2xHA-CR-*hip1* and SG200-CR-*hap1*-p*Pit2*::Hap1-2xHA-CR-*hip2*) as well as a double frameshift mutant for both hip1 and hip2 (SG200-CR-*hap1*-p*Pit2*::Hap1-2xHA-CR-*hip1-2*) using CRISPR-Cas9 mutagenesis in SG200-CR-*hap1*-p*Pit2*::Hap1-2xHA background strain. Maize seedlings were subsequently infected with the respective mutants, and SG200-p*Pit2*::SP-*mCherry*-HA strain was used as a control for Co-IP followed by MS analysis (Fig. 5a, mass-spectrometry results are summarized in Supplementary Data 12–14). In the samples infected with the Hap1-CR-*hip1* and Hap1-CR-*hip2* mutants, as well as in the Hap1-CR-*hip1/hip2* double mutant, Hap1 was unable to pull down SnRK1α, indicating that Hip1 and Hip2 influence Hap1-SnRK1α interaction.

To explore changes in protein interactions, proteins exclusively detected in frameshift mutants were subjected to GO enrichment analysis using PLAZA 5.0 and PPI analysis by the STRING database. GO and PPI results revealed significant association in 'cellular component organization or biogenesis' and 'Protein phosphatase activity' in Hap1-CR-hip1; 'protein modification', 'brassinosteroid-mediated signaling', 'starch metabolic process' in Hap1-CR-*hip2*; 'ER-Golgi transport,' 'calcium channel activity,' and 'hydrolase activity' in the Hap1-CR-*hip1/2* double frameshift mutant (Fig. 4b and Supplementary Fig. 5a–c).

In light of the observed analysis above, we focused on kinases and SnRK1 substrates, given that SnRK1α is the primary host target(s) of Hap1. Interestingly, all frameshift knockout mutants Hap1-CR-*hap1*, Hap1-CR-*hip1*, Hap1-CR-*hip1* and Hap1-CR-*hip1/2* interacted with phosphatases and trehalose-6-phosphate synthases, known SnRK1 substrates (Supplementary Data 15). Specifically, Hap1-CR-*hip1* interacted with poorly characterized proteins, such as protein phosphatase 2C (PP2C), a regulator of the signal transduction pathway involved in modulating receptor-like kinases and abscisic acid signaling[34], and phosphotyrosyl phosphatase activator (PTPA), which activates protein phosphatase 2A (PP2A) (Supplementary Data 15). Moreover, both Hap1-CR-*hip1* and Hap1-CR-*hip1/2* interacted with PPP2CB, the catalytic subunit of PP2A that negatively regulates cell cycle progression by controlling the timing and coordination of cell division[35]. In contrast, Hap1-CR-*hip2* interacted with enzymes involved in starch biosynthesis, as well as kinases and phosphatases associated with the brassinosteroid signaling pathway (Supplementary Data 15). Interestingly, both Hap1-CR-*hip2* and Hap1-CR-*hip1/2* mutants interacted with SnRK2 (Zm00001eb434400, Zm00001eb392580), a kinase involved in stress and abscisic acid (ABA)-mediated signaling pathways, as well as fructose-2,6-biphosphatases, which are important in regulating glycolysis and gluconeogenesis (Supplementary Data 15).

## Hip1 and Hip2 interact with ZmSnRK1α2 and influence the stability of the Hap1

To understand how Hip1 and Hip2 influence Hap1-SnRK1α interaction, we first investigated whether Hip1 and Hip2 directly interact with ZmSnRK1α. We performed a split-luciferase complementation assay in

$N.$ $benthamiana$, co-expressing Hip1$\Delta$SP-cLuc or Hip2$^{\Delta SP}$-cLuc with nLuc-ZmSnRK1α1, -ZmSnRK1α2, or -ZmSnRK1α3. Co-expression of Hip1$^{\Delta SP}$-cLuc and nLuc-ZmSnRK1α2 served as a positive control. Luminescence signal was observed only when Hip1$^{\Delta SP}$-cLuc or Hip2$^{\Delta SP}$-cLuc was co-expressed with nLuc-ZmSnRK1α2 (Fig. 6a–c), no signal

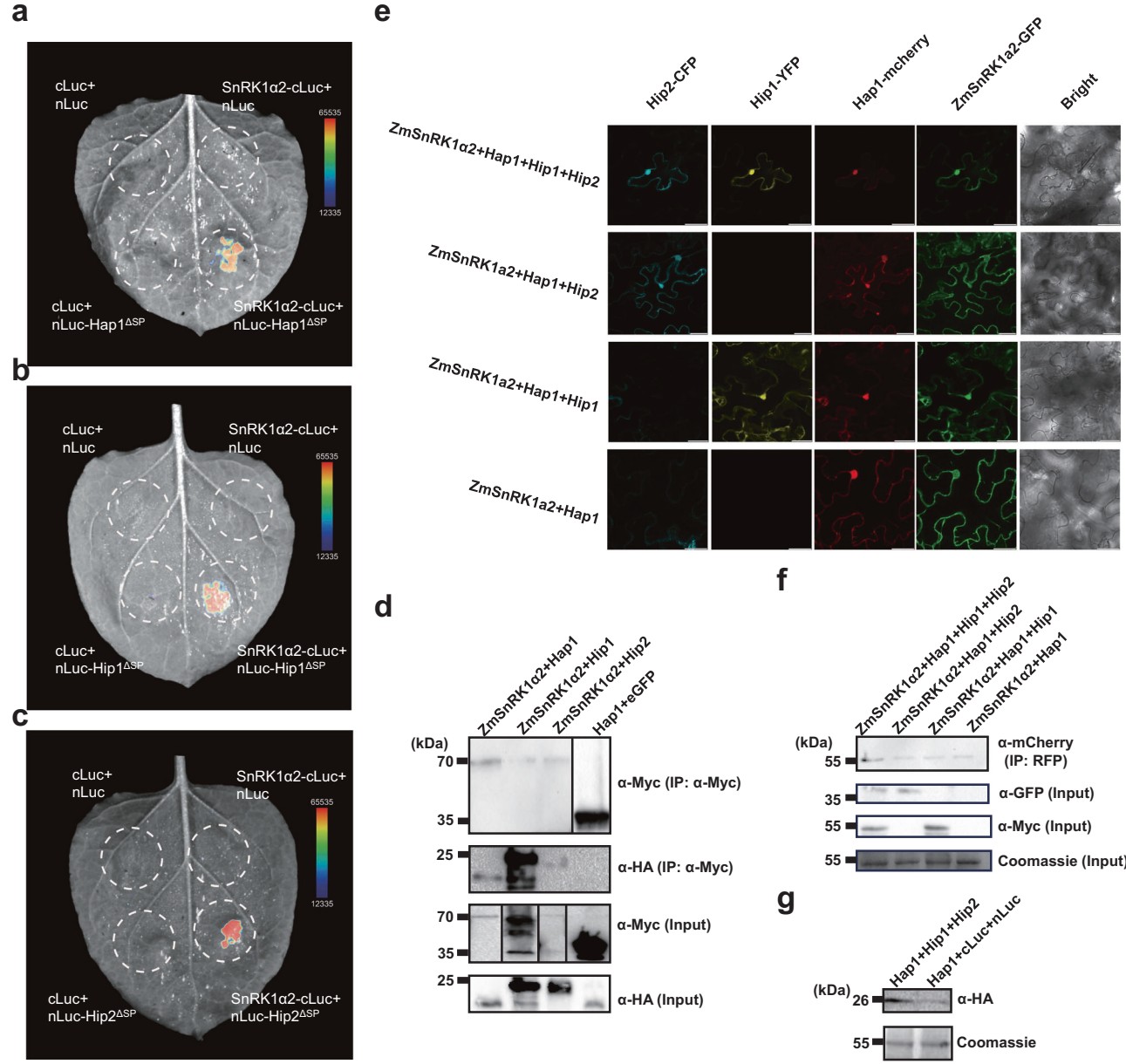

**Fig. 6 | Hip1 and Hip2 interact with ZmSnRK1α2 and influence the stability of Hap1. a–c** Split-luciferase complementation assay in $N.$ $benthamiana$. nLuc-Hap1$\Delta$SP or nLuc-Hip1$\Delta$SP or nLuc-Hip2$\Delta$SP or nLuc were co-expressed with ZmSnRK1α2-cLuc, or cluc. Luminescence was detected using Bio-Rad ChemiDoc™ and pseudo-fluorescence applied for enhanced visualization. **d** Co-immunoprecipitation (Co-IP) assay in $N.$ $benthamiana$. p2x35S-Hap1ΔSP-6xHA or p2x35S-Hip1ΔSP-6xHA or p2x35S-Hip2ΔSP-6xMyc were co-expressed with p2x35S- ZmSnRK1α2-4xMyc, or p2x35S-GFP-4xMyc. Proteins pulled down with Myc magnetic beads were detected using anti-HA or anti-Myc antibodies. Expected protein sizes: Hap1ΔSP-6xHA = 20.07 kDa; Hip1ΔSP-6xHA = 24.91 kDa; Hip2ΔSP-6xHA = 23.22 kDa; ZmSnRK1α2-4xMyc = 63.27 kDa; GFP-4xMyc = 31.8 kDa. **e** Co-localization of ZmSnRK1α2-GFP-4xmyc with Hap1-mCherry-6xHA, and/or Hip1-YFP-myc, and/or Hip2-CFP was assessed via $Agro$-$bacterium$-infiltration in $N.$ $benthamiana$ leaves. Images were captured at 3 days post-infiltration. All images include a 40 μm scale bar in the lower right corner. The GFP signal (ZmSnRK1α2) is shown in green, mCherry (Hap1) in red, YFP (Hip1) in yellow, and CFP (Hip2) in blue. Each image set includes individual fluorescence channels, a

bright-field image of the $N.$ $benthamiana$ leaf epidermis, and a merged image overlaying the fluorescence and bright-field signals. A 40 μm scale bar is shown in the lower right corner of each image. **f** Expression of co-expressed proteins. Western blots showing proteins Hap1-mCherry-6xHA, and/or Hip1-YFP-myc, and/or Hip2-CFP detected with anti-mCherry (Hap1), anti-GFP (Hip2), anti-myc (Hip1). Lanes correspond to: 1, co-expression of ZmSnRK1α2-GFP-4xmyc, Hap1-mCherry-6xHA, Hip1-YFP-myc and Hip2-CFP; 2, co-expression of ZmSnRK1α2-GFP-4xmyc, Hap1-mCherry-6xHA and Hip2-CFP; 3, co-expression of ZmSnRK1α2-GFP-4xmyc, Hap1-mCherry-6xHA and Hip1-YFP-myc; 4, co-expression of ZmSnRK1α2-GFP-4xmyc and Hap1-mCherry-6xHA. Molecular-weight markers (kDa) are shown to the left of each blot. Expected protein sizes: Hap1-mCherry-6xHA = 48.87 kDa; Hip1-YFP-4xmyc = 46.51 kDa; Hip2-CFP = 43 kDa. **g** Effect of Hip1 and Hip2 on protein accumulation of Hap1 upon co-expression in $N.$ $benthamiana$. Hap1ΔSP-6xHA was co-expressed with nLuc-Hip1 and Hip2-cLuc, co-expression with nLuc and cLuc served as the control. Western blots showing protein Hap1ΔSP-6xHA detected with anti-HA. Total protein loading was confirmed by Coomassie staining of the gel.

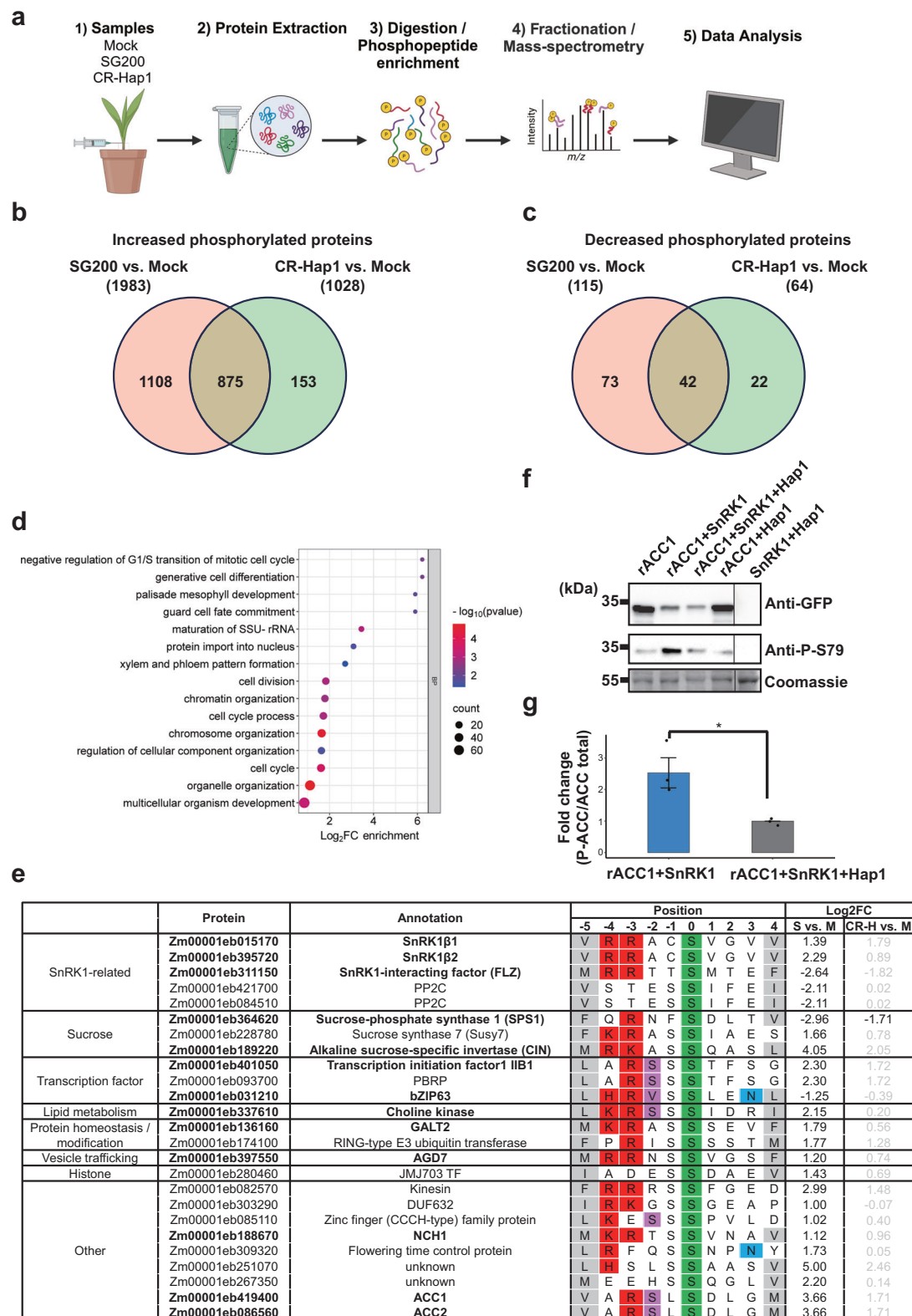

**Nature Communications** | (2026)17:1990

was detected for combinations with ZmSnRK1α1 or ZmSnRK1α3. Based on these results, we focused on the interaction between Hip1 or Hip2 and ZmSnRK1α2. To validate the interactions, we performed Co-IP assays. Hip1$^{\Delta SP}$-6xHA or Hip2$^{\Delta SP}$-6xHA were co-expressed with ZmSnRK1α2-4xMyc in *N. benthamiana* leaves. Immunoprecipitation using α-Myc pulled down Hip1$^{\Delta SP}$-6xHA and Hip2$^{\Delta SP}$-6xHA, but not in GFP-4xMyc control (Fig. 6d). Co-expression of Hap1$^{\Delta SP}$-6xHA and

ZmSnRK1α2-4xMyc served as a positive control and confirmed the interaction (Fig. 6d). These results show that both Hip1 and Hip2 interact with ZmSnRK1α2. In addition, we investigated the subcellular localization of Hap1, ZmSnRK1α2, Hip1, and Hip2 through co-agroinfiltration in *N. benthamiana*. Confocal microscopy demonstrated that all four proteins co-localized in both the cytoplasm and the nucleus (Fig. 6e). To ensure that the observed fluorescence signals

**Fig. 7 | Hap1 shapes host phosphorylation dynamics and SnRK1 signaling.**
**a** Schematic overview of the phosphoproteomics experiment: (1) seven-day-old maize seedlings were infected with mock, SG200, or CR-*hap1* and collected at 3 dpi. Created in BioRender. Doehlemann, G. (2026) https://BioRender.com/1zmsye4. (2) Total maize proteins were extracted and separated into total proteome and phosphoproteome fractions. (3) Protein peptides were phosphoenriched using titanium dioxide. (4) Peptides were fractionated and analyzed by mass spectrometry. (5) Identified spectra were mapped to the *Z. mays* genome to identify phosphorylation sites. **b** Venn diagram showing uniquely increased phosphorylated proteins in SG200 and CR-*hap1* compared to mock. **c** Venn diagram showing uniquely decreased phosphorylated proteins in SG200 and CR-*hap1* compared to mock. **d** GO enrichment analysis of proteins with increased phosphorylation in SG200 vs. CR-*hap1*. Significance is indicated by -Log10(*P*-value), with color shading from red (high significance) to blue (low significance). **e** Overview of the SnRK1-dependent consensus motif at P-5 and P + 4 relative to phosphorylated Ser/Thr residues. The color scheme follows the MEME Suite's coding for amino acids. Amino acid matching the known human AMPK consensus phosphorylation sequence is colored in pink and blue at P-2 and P + 3 positions, respectively. Thirteen proteins that perfectly match the SnRK1 consensus motif are indicated in bold.

When the motif criteria were relaxed to require only one of the hydrophobic residues at either P−5 or P + 4, a total of 25 proteins matched the SnRK1 motif. Log2FC values are color-coded: black for log2FC > 1 or < − 1 with *p* < 0.05, and gray for all others. **f** Hap1 inhibits SnRK1-dependent phosphorylation of rACC1. A GFP-fused 57-amino acid peptide surrounding the Ser79 phosphorylation site of rat acetyl-CoA carboxylase 1 (rACC1) was co-expressed with the indicated proteins in *N. benthamiana*. SnRK1 activity was assessed by detecting rACC1 phosphorylation at Ser79 (pS79 ACC) using a phospho-specific Anti-P-S79 ACC antibody. Total rACC1 levels were detected with an anti-GFP antibody. Lane 1: expression of rACC1 alone; Lane 2: co-expression of rACC1 and SnRK1; Lane 3: co-expression of rACC1, SnRK1 and Hap1; Lane 4: co-expression of rACC1 and Hap1; Lane 5: co-expression of Hap1 and SnRK1. The positions of the molecular weight ladder are shown on the right. The experiment was repeated three times with similar results. **g** Quantification of SnRK1-dependent rACC1 phosphorylation in the presence or absence of Hap1. The bar graph shows the fold change of P-S79 rACC1 normalized to total rACC1 (P-ACC/ACC-total). P-ACC/ACC-total ratios were scaled to the Lane 3 condition (set to 1) to obtain fold-change values. Data represent three biological replicates, each from an independent blot. Bars show mean ± SEM. Statistical significance was determined by a paired t-test on log2 transformed values.

originated from full-length proteins, we performed western blot analysis, which confirmed the expected protein sizes (Fig. 6f). Interestingly, when equal amounts of total protein were loaded, Hap1 accumulated higher when co-expressed with Hip1 and Hip2 (Fig. 6f). To test whether Hip1 and Hip2 influence the stability of Hap1, we co-expressed Hap1 with nLuc-Hip1 and Hip2-cLuc, using co-expression of Hap1 with nLuc and cLuc as control. Consistent with our initial observation, Hap1 accumulated to higher levels in the presence of Hip1 and Hip2 (Fig. 6g), suggesting that Hip1 and Hip2 increase the stability of Hap1.

## Hap1 shapes host phosphorylation dynamics and SnRK1 signaling

To access the role of Hap1 in host phosphorylation signaling during tumor formation, a quantitative phosphoproteomic analysis was conducted using mock-infected and *Ustilago*-infected (SG200 and CR-*hap1*) maize seedlings at 3dpi, with three biological replicates (Fig. 7a). Venn analysis, applying a cut-off of Log2FC ≥ 1 or ≤ − 1 cut-off, compared the phosphoproteome and total proteome to distinguish phosphorylation-specific changes. In SG200 vs. Mock (S vs. M), 3,618 phosphopeptides (2337 proteins) increased, and 180 phosphopeptides (148 proteins) decreased; total proteome analysis identified 1534 enriched and 745 depleted proteins (Supplementary Fig. 12a, b). Integration of these datasets yielded 1983 up- and 115 down-phosphorylated proteins (Fig. 7b, c). In Hap1-infected vs. mock (CR-H vs. M), 1,480 phosphopeptides (1,205 proteins) increased, and 117 (94 proteins) decreased; total proteome analysis found 1086 enriched and 567 depleted proteins (Supplementary Fig. 12c, d). Integrating these datasets resulted in 1028 up- and 64 down-phosphorylated proteins (Fig. 7b, c). Notably, SG200 induced twice as many phosphorylated proteins as CR-*hap1*, suggesting Hap1 is critical for targeting post-translational modifications. Total proteomic and phosphoproteomic data are summarized in Supplementary Data 16, 17.

To distinguish Hap1-dependent from Hap1-independent phosphorylation, we compared uniquely altered phosphorylated proteins in S vs. M and CR-H vs. M. In S vs. M, there were 1108 and 73 proteins with increased and decreased phosphopeptides; in CR-H vs. M, 153 increased and 22 decreased (Fig. 7b, c). Notably, 85% (875) of phosphorylated proteins in CR-H vs. M overlapped with those in S vs. M, suggesting largely shared signaling responses and minimal impact from loss of Hap1 (Fig. 7b, c). GO analysis of increased phosphoproteins in S vs. M revealed enrichment in processes like 'negative regulation of G1/S transition of mitotic cell cycle', 'peptidyl-serine modification', 'metabolic process regulation', 'autophosphorylation',

and 'gene expression regulation' (Supplementary Data 18). In CR-H vs. M, enriched terms included 'negative regulation of protein-containing complex disassembly' and 'response to stimulus' (Supplementary Data 19). Comparing SG200 and CR-*Hap1* (S vs. CR-H) identified 332 increased and 15 decreased phosphorylated proteins. GO terms enriched in SG200-infected leaf tissue included 'negative regulation of G1/S transition of mitotic cell cycle, 'cell division' and 'palisade mesophyll development (Fig. 7d). In SG200-infected tissue, increased phosphorylation was observed in several key proteins, including two retinoblastoma-related (RBR) proteins, RBR1 (Zm00001eb113470) and RBR3 (Zm00001eb037120) (Supplementary Data 17), which function as tumor suppressors by negatively regulating the cell cycle[36-38].

The data suggest that Hap1 promotes endoreduplication of mesophyll HTT cells by disrupting cell cycle regulation, potentially depending on carbon availability in tumors. Since SnRK1 is a key metabolic sensor linking energy status to transcriptional and post-translational responses, we investigated whether Hap1 affects host signaling through SnRK1-related pathways. We searched for known SnRK1 substrate motifs ($\phi$XXXX**S/T**XXX$\phi$) with specific residue preferences, including features from the human AMPK consensus motif[18,39]. Using the FIMO database, we scanned for these motifs in uniquely increased/decreased phosphoproteins from S vs. M (1108 increased,73 decreased), CR-H vs. M (153 increased and 22 decreased). Cross-referencing with phosphoproteomic data revealed 13 proteins perfectly matching the SnRK1 consensus sequence (Fig. 7e), and relaxing the motif criteria to include only one of the hydrophobic residuesP−5 or P + 4 identified in total 25 proteins matching the SnRK1 consensus sequence (Fig. 7e), though these may overlap with CDPK phosphorylation motifs and should be interpreted cautiously[18,40,41]. Notably, SnRK1α1, β1 and β2, and γ were highly phosphorylated in S vs. M, but not in CR-H vs. M, indicatingHap1 is required for specific phosphorylation events on SnRK1 subunits. Phosphorylation of SnRK1α1 occurred outside the canonical T-loop activation site, suggesting a regulatory role rather than direct activation.

To access Hap1-dependent metabolic reprogramming, we analyzed phosphorylation changes of known SnRK1 substrates and related enzymes. In S vs. M, increased phosphorylation was observed in sucrose synthase 2 (Susy 2) (Zm00001eb016290, Supplementary Data 17), Sucrose synthase 7 (Susy 7), and Alkaline sucrose-specific invertase (CINV), while decreased phosphorylation was observed in transcription factor bZIP63 and poorly characterized phosphatase C (PP2C) and FCS-like zinc finger (FLZ) (Fig. 7e). In S vs. CR-H, choline kinase, involved in lipid metabolism showed increased

phosphorylation, while decreased phosphorylation was observed in sucrose phosphate synthase 1 (SPS1), an enzyme involved in sucrose biosynthesis (Fig. 7e).

Next, we checked whether Hap1 influences SnRK1 activity. Using an adapted SnRK1 kinase activity assay[42], SnRK1 activity was assessed by Western blot detection of rat Acetyl CoA Carboxylase 1 (rACC1) phosphorylation following co-expression with a GFP-fused 57-amino acid peptide of rat acetyl-CoA carboxylase 1 (rACC1) in *N. benthamiana*. Co-expression of rACC1 with ZmSnRK1α2 resulted in strong phosphorylation of rACC1 (Fig. 7f). rACC1 alone or with Hap1 showed minimal phosphorylation, and Hap1 with ZmSnRK1α2 in the absence of rACC1 produced no detectable signal (Fig. 7f). In contrast, expression of rACC1 and ZmSnRK1α2 together with Hap1 led to a significantly reduced rACC1 phosphorylation (Fig. 7f, g). These results indicate that Hap1 suppresses SnRK1-dependent phosphorylation of rACC1 in vivo, suggesting an inhibitory effect of Hap1 on SnRK1 signaling activity.

## Discussion

This study reveals a virulence strategy employed by *U. maydis* to induce hypertrophy in host cells. Previously, Matei et al. observed that *U. maydis* effectors are deployed in a cell type specific manner, with distinct effectors inducing HTT and HPT, respectively[7]. This finding enabled the characterization of the HTT-specific effector Hap1. HTT-specificity is emphasized by Hap1's essential role in promoting the defining feature of HTT endoreduplication. In CR-*hap1*-infected tissues, the reduced nuclear size and the downregulation of cell cycle regulators compared to SG200 suggest that Hap1 promotes G1/S progression and endocycle entry in mesophyll cells, consistent with the role of endoreduplication[43]. In addition, phosphoproteomics reveals that Hap1 targets cell cycle control, RBR1/RBR3, key repressors of S-phase entry[36,37], were hyperphosphorylated in SG200 but not in CR-*hap1*, directly linking post-translational modification to HTT-specific cell cycle reprogramming. The *U. maydis*-induced starch accumulation is highly localized in infected tissues[7,11], with Matei et al.[7] reporting a significant increase specifically in HTT cells. Sosso et al. also observed that cells near the leaf vasculature undergo hypertrophy and retain starch granules longer than normal[11]. Given that cell expansion correlates with internal sugar concentrations[44], Matei et al.[7] proposed that the induction of hypertrophy in mesophyll cells is linked to increased starch accumulation. Our findings with the HTT-specific effector Hap1 further support this hypothesis, suggesting that starch accumulation may directly contribute to HTT. *U. maydis* infection affects leaf chloroplast dimorphism[7], causing starch granules to become less organized and more dispersed. The dispersed starch accumulation in infected leaves likely reflects subtle metabolic reprogramming induced by the pathogen. The significantly decreased starch accumulation in CR-*hap1* tissues, alongside transcriptional evidence for downregulation of starch biosynthetic genes (AGPase3, SBE I, and Ae1) and a shift toward alternative carbohydrate metabolism compared to SG200, indicates that Hap1 reprograms carbohydrate metabolism toward starch storage. This is further supported by our phosphoproteomics data, which show increased phosphorylation of starch precursor Susy[45] in the presence of Hap1.

We found that Hap1 interacts with ZmSnRK1α, a central metabolic sensor and kinase that integrates energy status with transcriptional and post-translational regulation in plants[15]. While the molecular mechanism by which the Hap1–SnRK1 interaction contributes to HTT formation remains to be elucidated, current evidence suggests that Hap1 facilitates host metabolic reprogramming by physically interacting with ZmSnRK1α (Fig. 8). SnRK1 is typically activated under low energy or pathogen attack[46], leading to the repression of anabolic metabolism such as starch synthesis (Fig. 8a), induction of catabolic processes, and activation of defense responses (Fig. 8a)[47]. This canonical SnRK1 response contrasts with the phenotype observed in Hap1-dependent HTT cells: increased starch accumulation. Our SnRK1 activity assay showed that Hap1 reduces SnRK1-mediated rACC1 phosphorylation, suggesting Hap1 may act as a modulator of SnRK1 signaling output., Furthermore, the transcriptomics data showed Hap1 represses SnRK1-triggered outputs once SnRK1 is activated, including ROS/ethylene defenses[48], ribosome biogenesis repression[49]. Phosphoproteomics data further substantiates SnRK1 subversion. SnRK1 complex subunits (α1, β1, β2, γ) exhibit Hap1-dependent phosphorylation in S vs. M (absent in CR-*hap1*), suggesting Hap1 is required for specific phosphorylation events on SnRK1 subunits, which likely alter complex function. Motif analysis identifies 13 proteins with perfect SnRK1 substrate motifs (φXXXXS/TXXXφ) uniquely phosphorylated in SG200. These phosphorylation data collectively indicate Hap1 interferes SnRK1 signaling to module host metabolic reprogramming.

Together, these findings support a model in which, upon infection, SG200 secretes Hap1, which inhibits SnRK1. This inhibition suppresses defense responses and prevents the repression of starch synthesis. SnRK1 targets phosphorylated due to Hap1 and Susy, involved in starch biosynthesis, are upregulated (Fig. 7b). In contrast, during CR-*hap1* infections, Hap1 is absent and can no longer inhibit ZmSnRK1α2. This leads to stronger defense responses and reduced starch accumulation. Meanwhile, sugar metabolites, may accumulate

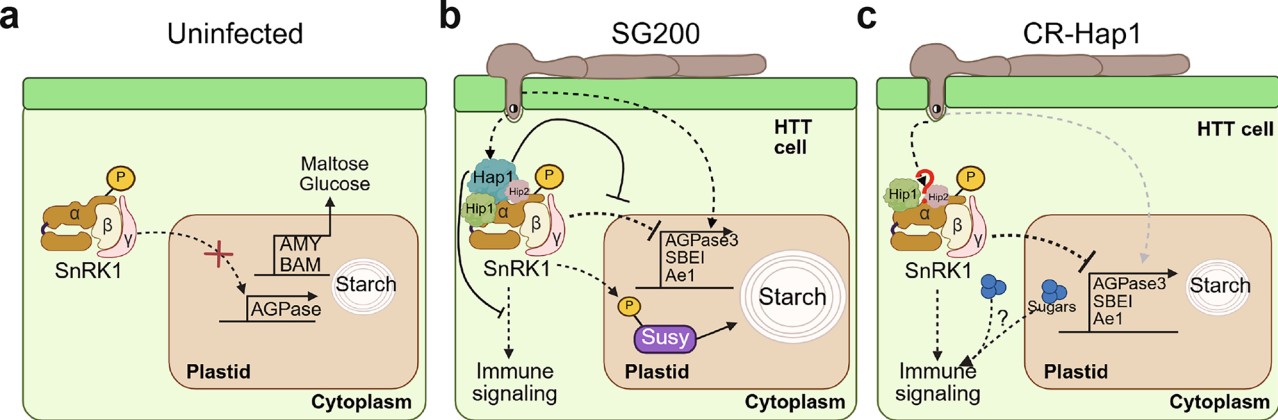

**Fig. 8 | Model depicting *U. maydis* promoting starch accumulation in hypertrophic tumor (HTT) cells via the Hap1- SnRK1 interaction. a** Activation of SnRK1 inhibits energy-consuming processes (e.g., starch synthesis) under normal conditions. **b** Hap1 inhibits SnRK1 to promote starch accumulation during *U. maydis* infection. **c** The absence of Hap1 affects starch accumulation during *U. maydis* infection. Created in BioRender. Doehlemann, G. (2026) https://BioRender.com/ldsm25g.

and act as signals to trigger immune responses (Fig. 7c). Intriguingly, Hap1 operates within a multi-effector module to interact with SnRK1 (Fig. 7b). Hap1 stability is enhanced by two Hap1 interacting effectors, Hip1 and Hip2. Loss of Hip1/Hip2 disrupted Hap1-SnRK1 interaction and alters the Hap1 interactome.

Plant pathogens often deploy multiple effectors against a single host target[50], sometimes in temporally distinct "waves" during colonization[51]. However, Hap1, Hip1, and Hip2 show highly similar expression patterns peaking from 2 dpi in HTT cells, suggesting synchronous deployment. Our observations indicate that Hap1, Hip1, and Hip2 interact with each other and with ZmSnRK1α2, and that Hip1 and Hip2 can enhance the stability of Hap1 and likely its interaction with ZmSnRK1α2, suggest that these proteins may form a complex. However, the current state of knowledge leaves open the question of whether a functional protein complex of the three effectors and the common target ZmSnRK1α2 actually exists within the host cell. Therefore, elucidating the structure of the protein interactions described here will be a major goal of our further research. In another line, generating maize ZmSnRK1α2 mutants, which could be tested for their response to *U. maydis* infection, would be critical to determining the role of this interaction in tumor formation. Ultimately, future research aims to elucidate the mechanistic relationship between pathogen-induced tumor formation, metabolic reprogramming, and immunosuppression.

## Methods

### Strains, fungal, and plant growth conditions
Plasmids were cloned in *Escherichia coli* Top10 strains. For transient protein expression in *N. benthamiana*, *Agrobacterium tumefaciens* GV3101 was used. *E. coli* and *A. tumefaciens* were grown in dYT-liquid medium (1.6% w/v peptone, 1% w/v yeast extract and 0.5% w/v NaCl) or YT agar plates with appropriate antibiotics at 37 °C with shaking at 200 rpm or 28 °C, respectively. *U. maydis* was grown in YEPS_Light liquid medium (0.4% w/v yeast extract, 0.4% w/v peptone and 2% w/v sucrose) or on PD-agar plates at 28 °C with shaking at 200 rpm. *Zea mays* Golden Bantam (GB) or Early Golden Bantam (EGB) plants were grown under controlled glasshouse or phytochamber conditions (16 h light at 28 °C and 8 h dark at 22 °C). *N. benthamiana* plants were cultivated in a glasshouse (16 h light and 8 h dark at 22 °C).

Mutants were generated in *U. maydis* SG200 solopathogenic or effector mutant strains using CRISPR mutagenesis as previously described[52]. For complementation, the p123 plasmid[53] containing an *ip* allele for carboxin (*cbx*) resistance[54] was linearized with SspI or AgeI and integrated into the genome via homologous recombination. Single copy integration was confirmed by Southern blot (data not shown). Primer details are in Supplementary Data 20.

### Maize infection, and disease scoring
Seven-day-old maize seedlings were inoculated with *U. maydis* ($OD_{600}$ = 1 for disease scoring and $OD_{600}$ = 3 with 0.1% tween-20 for microscopy, (interactome/total/phospho) proteomics, and RNA-seq). Disease scoring was performed at 12 days post-inoculation (dpi) as previously described[55], the scoring scale is provided in Supplementary Fig. 13. Disease indexes were assigned as follows: 9 for dead plants, 7 for heavy tumors, 5 for tumors, 3 for small tumors, 1 for chlorosis, and 0 for no symptoms. The disease index was used for assessing statistical significance using Student's *t* test with three independent biological replicates. Disease scoring data of maize infection is provided in Source Data.

### Leaf staining, tissue embedding, sectioning and microscopy
For sectioning, leaves were harvested at 6dpi and cut into approximately 2 cm (2 cm below the infection site). Tissues were embedded as previously described[7], molded in Peel-A-Way™, and sectioned transversely into 13 μm slices. For staining, sections were treated with

Lugol's iodide (IKI) solution (Roth) for starch staining and propidium iodide (Sigma-Aldrich) for nuclei staining. Imaging was performed using a Nikon Eclipse Ti inverted microscope with Nikon Instruments NIS-ELEMENTS software (561 nm excitation and at 590–603 nm emission for propidium iodide) or a Thunder microscope with Leica imaging LAS X software. Nuclear size was measured as previously described[7].

### RNA preparation and RNA-Seq analysis
For total RNA extraction, leaves were harvested at 3dpi and homogenized in liquid nitrogen using TRIzol Reagent (Invitrogen) followed by DNA removal with the Turbo DNA-Free™ Kit (Ambion Life Technologies™). Three independent biological replicates were subjected to 150-bp paired-end RNA-seq on Illumina NovaSeq 6000 (Illumina) at Novogene (Cambridge, UK). RNA-seq analysis was performed as previously described[56].

### GO enrichment and protein-protein interactions (PPI) analysis
GO enrichment analysis was performed using Plaza 5.0[57] with Fisher's exact test with Bonferroni correction (*P*-value < 0.05). Results were grouped into upper hierarchical parent terms using REVIGO PPI analysis was performed using the STRING database[58].

### Co-immunoprecipitation (Co-IP) in maize and sample preparation
Infected leaves were harvested at 3 dpi. Leaf sections of ~ 4 cm were cut 1 cm below the infection site and immediately frozen in liquid nitrogen. Frozen tissue was ground to a fine powder. 3 independent biological replicates were prepared. Each biological replicate consisted of a pool of at least 10 plant leaves. The SG200 strain was used as the control for infection. For protein extraction, 1 ml of powdered leaves was mixed with 1 ml of lysis buffer (50 mM Tris–HCl pH 7.5, 150 mM NaCl, 2 mM EDTA, 10% glycerol, 1% Triton X-100, 5 mM DTT and cOmplete™ protease inhibitor (Roche)). The mixture was incubated on ice for 30 min and centrifuged twice at 4 °C with $13\,000 \times g$ for 30 min to obtain protein supernatant. 1 ml of the protein supernatant was incubated with anti-Myc or anti-GFP (ChromoTek) magnetic beads at 4 °C, 1-2 h with rotation, and washed at least five times. The washed beads were either subjected to mass spectrometry analysis or resuspended in 80 μl of 2 × SDS-loading buffer and boiled at 99 °C for Western blot analysis. Detection was performed using anti-GFP (Roche, 11814460001, Dilution: 1:1000), anti-Myc (Invitrogen, R951-25, Dilution: 1:1000), or anti-HA (Sigma-Aldrich, 12158167001, Dilution: 1:2000) antibodies.

For sample preparation, LC-MS/MS data acquisition, and data analysis of interactome proteomics was performed as previously described for HA enrichment pull-down[59]. Enriched proteins were digested on-bead using trypsin. Briefly, beads were incubated with digestion buffer (50 mM Tris pH 7.5, 2 M urea, 1 mM DTT, 5 ng/μL trypsin) for 30 min at 30 °C, followed by collection of the supernatant. A second digestion buffer (50 mM Tris pH 7.5, 2 M urea, 5 mM CAA) was added, and supernatants were combined and incubated overnight at 32 °C. Digestion was stopped with TFA, and peptides were desalted using C18 StageTips[60]. Peptides were analyzed on an EASY-nLC 1200 coupled to a Q Exactive Plus (Thermo Fisher). Peptides were separated on a 16 cm in-house packed C18 column using a 115 min linear gradient (5–95% ACN, 0.1% FA) at 300 nL/min. MS spectra were acquired in data-dependent TOP15 mode, with MS1 at 70,000 FWHM and MS2 at 17,500 FWHM. Precursors with charge +1, > 6, or unassigned were excluded; dynamic exclusion was 30 s.

### Sample preparation for total proteome and phosphoproteome
Proteins were extracted with 1 ml extraction buffer (8 M urea, 20 μl/ml phosphatase Inhibitors (Sigma, P5726-5ML and P0044-5ML), 5 mM DTT) and incubated for 30 min with shaking. Samples were alkylated

with 14 mM CAA and quenched with 5 mM DTT. 500 μg of total proteins were diluted to 1 M urea in 100 mM Tris-HCl, pH 8.5, 1 mM CaCl$_2$, and digested with 5 μg LysC (WAKO) in 50 mM NH$_4$HCO$_3$ for 4 h at RT. 5 μg trypsin was added to samples and diluted with 100 mM Tris-HCl, pH 8.5, 1 mM CaCl$_2$. Samples were mixed and incubated overnight at 37 °C. Digests were acidified with TFA to 0.5% and desalted using C18 SepPaks (1cc cartridge, 100 mg (WAT023590)). Samples were eluted with 80% ACN/0.1% TFA and dried.

For library samples, 5 μL aliquots of samples were pooled and fractionated using SCX StageTips packed with Empore Cation SPE disks (Empore Cation 2251 material). Samples were fractionated with ammonium acetate gradient (25–500 mM in 20% ACN, 0.5 % FA) into nine fractions, with additional elutions with 1% ammonium hydroxide, 80% ACN and 5% ammonium hydroxide, 80% ACN. Fractions were eluted by centrifugation (5 min, 500 x $g$), dried, and resuspended in 10 μl A* buffer. For DDA analysis, the remaining 40 μl eluted peptides from the SepPack purification were dried and resuspended in 10 μl A* buffer. Peptide concentration was determined by Nanodrop, and samples were diluted to 0.1 μg/μL for measurement.

For phosphopeptide enrichment by metal-oxide chromatography (MOC) (adapted from Nakagami, 2014), peptides were evaporated to 50 μL and diluted with sample buffer (2 ml ACN, 820 μl lactic acid (LA), 2.5 μl TFA / 80% ACN, 0.1% TFA, 300 mg/ml LA). MOC tips were loaded with 3 mg TiO$_2$ beads (Titansphere TiO$_2$, 10 μm (GL Science Inc, Japan, Cat. No. 5020-75010)) in 100 μL MeOH in a C8 micro column. MOC tips were placed onto a 96-well plate (Protein LoBind, (Eppendorf Cat. No. 0030504100) and equilibrated. Samples were loaded onto the tips, centrifuged (10 min at 1000 × $g$), and reloaded for second centrifugation. Tips were washed using solution C and B with centrifugation (1500 × $g$, 5 min) and transferred to a fresh 96-well plate containing 100 μl of 20% phosphoric acid for elution of enriched phosphopeptides. Peptide was eluted sequentially with 50 μl elution buffer 1 (5% NH$_4$OH) and buffer 2 (10% piperidine), desalted using C18 StageTips [98], dried in a vacuum evaporator, and dissolved in 10 μl A* buffer for MS analysis. This experiment was performed with 3 biological replicates.LC-MS/MS Acquisition

For Co-IP samples, peptides were analyzed on an EASY-nLC 1200 (Thermo Fisher) coupled to a Q Exactive Plus mass spectrometer (Thermo Fisher). Peptides were separated on a 16 cm in-house packed C18 column (75 μm ID) using a 115 min linear gradient of 5–95% acetonitrile (ACN) in 0.1% formic acid (FA) at 300 nL/min. Mass spectra were acquired in data-dependent TOP15 mode. MS1 scans were recorded at 70,000 FWHM over 300–1750 m/z and a target value of $3 × 10^6$ ions.; MS2 scans were acquired at a resolution of 17,500 FWHM with a target value of $10^5$ ions, a maximum injection time (max.) of 55 ms and a fixed first mass of m/z 100. Precursors with charge + 1, > 6, or unassigned were excluded, and dynamic exclusion was set to 30 s.

For total proteome, library, and phosphoproteome samples, peptides were analyzed on an Ultimate 3000 RSLC nano (Thermo Fisher) coupled to an Orbitrap Exploris 480 equipped mass spectrometer with a FAIMS Pro interface (Thermo Fisher). Peptides were pre-concentrated on an Acclaim PepMap 100 pre-column (75 μm × 2 cm, C18, 3 μm, 100 Å, Thermo Fisher) at 7 μL/min for 5 min. Separation was performed on a 16 cm fritless silica emitter (75 μm ID, New Objective, in-house packed with ReproSil-Pur C18 AQ 1.9 μm resin) using a segmented linear gradient of 5–95% solvent B (ACN, 0.1% FA) over 115–130 min at 300 nL/min. MS spectra were acquired in data-dependent TOP_S mode with a 2 s cycle time. MS1 scans were acquired with a mass range of 320–1200 m/z at a resolution of 60,000 FWHM and a normalized AGC target of 300%, and MS2 scans with a target value of 75% ions at a resolution of 15,000 FWHM, at an automated injection time and a fixed first mass of m/z 100. For FAIMS, compensation voltages of − 45/− 60 V (total proteome and library) or − 45/− 65 V (phosphoproteome) were applied. Precursors with charge + 1, > 6, or unassigned were excluded; dynamic exclusion was 30–40 s.

## Data analysis

Raw data were processed using MaxQuant (v.1.6.3.4, http://www.maxquant.org/) with label-free quantification (LFQ) and iBAQ enabled [99,100]. For co-immunoprecipitation (Co-IP) experiments, MS/MS spectra were searched against a combined protein database containing Zea mays (v5_NAM), Ustilago maydis (UniProt), sequences of tagged effector proteins and corresponding controls, 248 common contaminant proteins, and reverse decoy sequences. For total proteome and phosphoproteome analyses, spectra were searched against Z. mays (v5_NAM), common contaminants, and decoy sequences. Trypsin specificity was required, allowing up to two missed cleavages. The minimum peptide length was set to seven amino acids. Carbamidomethylation of cysteine residues was specified as a fixed modification, while oxidation of methionine and protein N-terminal acetylation were included as variable modifications. For phosphoproteomics, phosphorylation of serine, threonine, and tyrosine residues was included as an additional variable modification. Peptide-spectrum matches (PSMs), peptides, and proteins were filtered at a false discovery rate (FDR) of 1% using a target–decoy strategy. The "match between runs" option was enabled for total proteome and phosphoproteome analyses, with library samples set to "match from" and DDA samples to "match from and to" in group-specific parameters.

For Co-IP experiments, searches were performed in a binary manner (sample versus control). MaxLFQ values were analyzed using Perseus (v1.5.8.5). Reverse hits and proteins identified only by site were removed, and LFQ intensities were log2 transformed. Proteins were retained for statistical testing if at least two valid values were present in one condition. Two-sample Student's $t$ tests were performed using a permutation-based FDR of 5%. Alternatively, a more stringent filtering requiring three valid values in one condition was applied, followed by imputation of missing values from a normal distribution (1.8 downshift, column-wise). Volcano plots were generated in Perseus using an FDR of 5% and an S0 value of 1. Results were exported for further processing in Excel.

For deep total proteome analysis, statistical processing was performed in Perseus (v1.6.14.0). Quantified proteins were filtered to remove reverse hits and proteins identified only by site. MaxLFQ values were log2 transformed and samples grouped by condition. Proteins with at least three valid values in one condition were retained. To account for different types of missing values, data were separated into missing-at-random (MAR) and missing-not-at-random (MNAR) datasets based on the presence of at least one valid value per group (Lazar et al., J. Proteome Res. 2016). Missing values were imputed using the imputeLCMD R package integrated in Perseus: MAR values were imputed using a k-nearest neighbor approach (KNN, $n = 5$), while MNAR values were imputed using the MinProb method ($q = 0.01$, tune; sigma = 1). After merging imputed datasets, two-sample Student's $t$ tests were performed with a permutation-based FDR of 5%. Results were exported for downstream analysis.

Phosphoproteomics data were analyzed at the phosphopeptide level using intensities from the "modificationSpecificPeptides" output in Perseus (v1.6.14.10). Reverse hits and contaminant entries were removed, and only phospho-modified peptides were retained. Intensities were log2-transformed and grouped by condition. Imputation and statistical testing were performed as described for the total proteome dataset, including MAR/MNAR separation, KNN and MinProb imputation, and two-sample Student's t-tests with permutation-based FDR correction.

## Nuclear and cytoplasmic extraction

Maize leaves were collected at 3dpi following infection with the indicated strain (OD$_{600}$ = 3), frozen in liquid nitrogen, and ground to a fine powder. Subcellular fractionation was performed based on a published method [61], with the procedure described in detail below. The powdered

tissue was resuspended in 2 ml cold Lysis Buffer (20 mM Tris-HCl (pH 7.4), 25% glycerol, 20 mM KCl, 2 mM EDTA 0.4 ml, 2.5 mM MgCl2, 250 mM sucrose, 5 mM DTT and cOmplete™ protease inhibitor (Roche)) and homogenized by gentle shaking or pipetting. The homogenate was filtered sequentially through a 40 µm nylon meshes to remove debris. Nuclei were pelleted by centrifugation at $1500 \times g$ for 10 min at 4 °C, and the supernatant was collected as the cytoplasmic fraction. The nuclear pellet was washed by resuspension in 3 ml NRBT (20 mM Tris-HCl (pH 7.4), 25% glycerol, 2.5 mM MgCl2, 0.2% Triton X-100), followed by centrifugation under the same conditions. This wash was repeated twice. Nuclei were then resuspended in 3 ml NRB (20 mM Tris-HCl (pH 7.4), 25% Glycerol, 2.5 mM MgCl2) and centrifuged to remove Triton X-100. Nuclei were resuspended in 400 µl NSB (20 mM Tris-HCl (pH 7.4), 25% Glycerol, 2.5 mM MgCl2, 15.1% (w/v) Sucrose), flash-frozen in liquid nitrogen, and stored at − 80 °C. The anti-HA antibody were used for detection by using the ChecmiDoc MP machine (BioRad).

### Split Luciferase complementary assay

Hap1, SnRK1α1-3, Hip1 and Hip2 coding sequences were fused to the N-terminal (nLUC) or C-terminal (cLUC) halves of luciferase and cloned into binary vector pCAMBIA1300 and subsequently introduced into *Agrobacterium tumefaciens* strain GV3101. The primers used for cloning are listed in Supplementary Data 20. For transient expression, 4–5-week-old *N. benthamiana* leaves were infiltrated with Agrobacterium suspensions carrying the gene of interest, mixed with p19 at $OD_{600} = 1$ in infiltration buffer (10 mM MgCl₂, 10 mM MES, 100 µM acetosyringone). 2-3 days post-infiltration, leaves were sprayed with 1 mM D-luciferin (Promega) and incubated in the dark for 10 min. Luminescence was detected using a ChemiDoc MP machine (Bio-Rad) from at least three independent plants.

### Co-IP and subcellular localizationin N. benthamiana

Using the primers listed in Supplementary Data 20, *Hap1*, *SnRK1α1-3*, *Hip1* and *Hip2* were cloned into MoClo level-0 plasmid pAGM1287. These constructs were then assembled with the desired tags into a level-1 binary vector and transformed into *A. tumefaciens* strain GV3101. Agroinfiltration and Co-IP were done as described above. Tagged constructs were introduced into *A. tumefaciens* strain GV3101. Subcellular localization of the expressed proteins was observed using a confocal laser scanning microscope (Leica, TCS SP8). At least three independent leaves were imaged per construct.

### SnRK1 activity assay

The SnRK1 activity assay was performed following the previously described assay[42]. A 57-amino acid peptide surrounding the Ser79 phosphorylation site of rat acetyl-CoA carboxylase 1 (rACC1) was fused to GFP. The plasmid rACC1-GFP was kindly provided by Prof. Dr. Wolfgang Dröge-Laser (Pharmaceutical Biologie, University of Würzburg). SnRK1 activity was assessed by Western blot detection of rACC1 phosphorylation following co-expression with rACC1 in *N. benthamiana*. After agroinfiltration, tissue from multiple infiltrated leaves (from three plants per pool) was collected and pooled to produce one biological replicate. Three replicates (each a separate pool prepared) were performed. Phospho-Acetyl-CoA Carboxylase (Ser79) (anti-P-S79 ACC) antibody (CST, #3661, Dilution: 1:1000) was used to detect phosphorylation of ACC at serine 79. Protein extracts were separated by SDS–PAGE and probed with anti-P-S79 and anti-GFP (total ACC–GFP). Band intensities were quantified in ImageJ. For each replicate, the p-ACC signal was background-subtracted and normalized to total ACC–GFP from the same lane. P-ACC/ACC-total ratios were normalized within each blot to a designated sample, which was assigned a value of 1. Normalized values were $\log_2$-transformed and compared between groups using a paired t-test.

### Reporting summary

Further information on research design is available in the Nature Portfolio Reporting Summary linked to this article.

### Data availability

The raw RNA-seq data generated in this study have been deposited in the NCBI Gene Expression Omnibus database under accession code GSE282754. The processed RNA-seq data, including differential expression results (Supplementary Data 1–3) and Gene Ontology (GO) enrichment analysis results (Supplementary Data 4–6), are available in the Supplementary Data file. The mass spectrometry proteomics data used in this study are available in the ProteomeXchange Consortium database via the PRIDE repository[62] under accession code PXD057668 (interaction proteome) [https://www.ebi.ac.uk/pride/archive/projects/PXD057668] and PXD057676 (total and phosphoproteome) [https://www.ebi.ac.uk/pride/archive/projects/PXD057676]. Additional information can be requested from the corresponding author (GD) upon reasonable request. Source data are provided in this paper.

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

## Acknowledgements

We express our gratitude to Anne Harzen for her valuable assistance in MS sample processing. We are grateful to Prof. Dr. Wolfgang Dröge-Laser for kindly providing the rACC1 plasmid and for recommending the SnRK1 activity assay. We acknowledge the funding received from the European Research Council (ERC) under the European Union's Horizon 2020 research and innovation program (grant agreement No 771035). We received support from the Deutsche Forschungsgemeinschaft (DFG, German Research Foundation) under Germany's Excellence Strategy-EXC-2048/1- Project ID: 390686111. G.D. and Y.J.L. acknowledge funding by the DFG through project DO1421/3-3. The Max Planck Society is acknowledged for the financial support of S.C.S. and H.N.

## Author contributions

G.D. and Y.J.L. designed the research. Y.J.L., D.Z., and M.E. conducted the experiments. G.S. and D.Z. performed bioinformatics analysis of RNA-Seq data. S.C.S. and H.N. performed the MS and MS data analysis. Y.J.L., G.D., and D.Z. wrote the paper with contributions from the other authors.

## Funding

## Competing interests

The authors declare no competing interests.
