## [Transparent Peer Review file · Nature Communications]

Ustilago maydis disrupts carbohydrate signaling networks to induce hypertrophy in host cells

Corresponding Author: Professor Gunther Doehlemann

Version 0:

Reviewer comments:

Reviewer #1

(Remarks to the Author)

In this manuscript, the authors focus on three effectors, Hap1-3 (hypertrophy-associated proteins), which were identified as virulence factors promoting hypertrophic mesophyll tumor cells (HTT), with Hap1 identified as a key virulence factor. Hap proteins form protein complex, and Hap1 interacts with maize SnRK1, disrupting energy regulation. Transcriptomic, proteomics and phosphoproteomics show that Hap1 targets SnRK1, reprogramming host metabolism and energy for hypertrophy. The topic is very interesting. However, some of the conclusions are not well supported by the data, especially the relationship between energy metabolism and Ustilago maydis is mostly based on omic-data analysis, and lacks solid gene functional study.

The authors used 2xHA-complemented strains for IP and subsequent mass-spectrometry assay. The authors stated that they used infected maize leaves for protein extraction, which contain both the pathogen and plant cells. Based on the mass results, the author concluded that Hap proteins form complex in planta. Since this is a mixed tissue, the identified candidate interacting proteins can not be simply concluded that they interact in planta, although it was validated that Hap protein can indeed interact with each other. Maybe Hap effectors only interact inside U. maydis and do not interact in plant cells. This need to be addressed. If Hap proteins form complex in planta, what is the meaning of Hap effector interacting with each other inside plant cells?

Following up the above questions, does Hap2 protein interact with Hap1 to form homo-dimers?

The authors claim that Hap1 is the dominant effector based on Figure S3, however, based on Figure 1b, all three Hap genes are required for full virulence. Does the authors compare the virulence of Hap2 and Hap3 single mutant with the Hap1/2/3 triple mutant?

The authors states that Hap1 interact with SnRK1 α 2 and SnRK1 α 3 (line 171-173), but no data was shown for Hap1 and SnRK1 α 3 (Fig .3a,b). Based on Fig 3c, Hap1 also interact with SnRK1 α 1? Why the interaction is inconsistent by different interaction systems?

In Figure S4, SnRK1 α 4 is also there, why the authors did not test the interaction between Hap1 and SnRK1 α 4?

Does Hap2 and Hap3 also interact with SnRK1?

Protein interaction confirmation using only LCI and Co-IP is not enough. The authors need to do BiFC / Pull-down or yeast two hybrid assay to confirm the interactions.

The authors did a lot of omic analysis, such as RNA-seq, proteomic and phosphoproteomic, and try to make solid conclusion from the omic data analysis. Although the data analysis seems support the conclusion perfectly, however, this is simply too ambitious and need point-to-point validations. The authors need to focus on key genes and make gene mutant of that gene(s) and functionally validate it (the author need to test disease phenotypes of target gene mutants). This is a major issue of this manuscript.

Reviewer #2

(Remarks to the Author)

The manuscript describes how *Ustilago* disrupts the carbohydrate signaling network to induce hypertrophy in host cells. They identified three hap effector molecules and demonstrate that Hap1 potentially targets Snf1-related kinase 1 (SnRK1) and disrupts T6P-SnRK antagonistic relationship to regulate cell cycle and starch biosynthesis in the host.

The manuscript is mostly descriptive and based on proteomics and transcriptomics data. Although it demonstrates the interaction between the effector and the target host proteins, but the relevance of the interaction with loss and gain of functions studies is missing. Following are some of the major concerns about the manuscript.

1. The English language is not clear, and difficult to understand. The manuscript should be simplified by focusing on the physiological relevance of Hap1 interaction with SnRK1 and how this affects T6P function to modulate starch metabolism. Also, how hypertrophy is modulated during Hap1-SnRK1 interaction should be clearly demonstrated.
2. The rationale of selecting Hap1-3 as a candidate effector paralogs is not clear. The table 1 enlists of some of the effectors, but not clear on what basis these genes were selected from the list reported in Rizzi et al and Zuo et al. The data related to knockout and complementation, with respect to the pathogenesis of *Ustilago* for each of the characterized effector genes, should be provided.
3. The biochemical functions of Hap1-3 is not described in the manuscript. Also, the phylogenetic analysis and evolutionary conservation of these genes should be briefly described, for better understanding.
4. The physiological relevance of Hap effector's interaction with each other in planta, is not clear. Do they interact in a complex with the target host proteins? It would be more relevant to focus on unraveling the common interacting partners of Hap1 paralogs in the host.
5. Have you tested whether Hap2 or Hap3 interact with the SnRK1 protein in planta. In lines, 199-200 on page 8, it is mentioned that knockout mutants of Hap1 interact with the SnRK substrate, this needs to be properly described.
6. The role of Hap1 in modulating starch reprogramming and endoreduplication is mostly obtained from the transcriptome analysis. The direct involvement of Hap1 in regulating these processes is not clear. It is possible the observed changes may be due to the pleiotropic effect of SnRK1 modulation.
7. It will be helpful if the authors test the direct involvement of Hap1 protein in regulating the cellular processes in the host. In this regard, creating a modified Hap1 wherein the active site is mutated and testing the effect of the WT and variants on the SnRK1 activity and T6P functions should be helpful.
8. Whether Hap1 interferes with the biochemical function of SnRK1 or T6P? The pathological development of *Ustilago* on the SnRK1 and T6S mutants should be tested.

Reviewer #3

(Remarks to the Author)

The authors demonstrated that Hap1, Hap2, and Hap3 form a protein complex, with only Hap1 shown to interact with maize ZmSnRK1 α , its primary target. The triple hap deletion mutant displayed a virulence phenotype similar to the single hap mutants, suggesting that all three HAP proteins are equally important in the effector complex formation. However, the biological relevance of the complex formation and its interaction with ZmSnRK1 α has not been investigated in this study. Additionally, the authors stated that Hap1 prevents SnRK1 inhibition by high levels of trehalose-6-phosphate (T6P), but this aspect was also not explored in this work.

The impact of the HAP complex on the regulation of SnRK1 activity remains unclear, despite the authors' efforts to investigate it using proteomics, transcriptomics, and phosphoproteomics.

Based on the GO and PPI results, there are both similarities and differences in the GO terms identified for the individual mutants, which may suggest functional similarities and differences among the HAP proteins. Unfortunately, the manuscript does not investigate whether Hap1 has a unique function distinct from Hap2 and Hap3. If all three HAP proteins share the same function—forming a complex to regulate SnRK1—I would expect to see similar outcomes in their phosphoproteomics studies and the starch accumulation levels in the single hap deletion mutants.

ZmSnRK1 is reported to play a defensive role; however, its mechanism of defending against pathogens by mobilizing energy remains unclear. While the authors demonstrated reduced starch accumulation in the hap1 mutant, no direct connection between Hap1 and SnRK1 has been established. The authors should address this by investigating starch accumulation patterns in SnRK1 deletion or overexpression transgenic lines.

- What is the significance of forming the Hap effector complex in interacting with ZmSnRK1 α or regulating the ZmSnRK1 α activity?
- Can the interaction of HAP1 affect the phosphorylation of SnRK1?
- Can Hap2 or Hap3 interact with ZmSnRK1 α ?
- There is no evidence provided to demonstrate that Hap1 prevents SnRK1 inhibition by high levels of trehalose-6-phosphate (T6P)
- It is known that ZmSnRK1 can form a complex, does the interaction with HAP1 disrupt the ZmSnRK1 complex or does HAP1 simply prevent ZmSnRK1 inhibition by T6P?
- ZmSnRK1 localizes to both nucleus and cytoplasm. Where does the interaction with HAP1 take place?

This information is crucial for elucidating the mechanism by which the Hap complex regulates HTT formation for fungal virulence, but it is missing.

Other comments:

Line 154-155: The three effectors form a complex, which is important for their virulence function. The similar virulence phenotype in the triple and single mutants did not support that Hap1 has a dominant role.

Line 165: Please provide the accession numbers for ZmSnRK1 α subunits in the main text.

Line 170-171: The split-luciferase complementation assay for the interaction of Hap1 and ZmSnRK1 α 3 is missing.

Fig 3c: Please show that the ZmSnRK1 α 1-3 proteins are not non-specifically bound to the Myc-beads.

Line 181-215 and Fig. 4: I don't understand how this analysis can support that the effector complex is important for orchestrating a regulatory cascade and mediating interactions between Hap1 and ZmSnRK1 α , since they have different GO terms. The data retrieved from the LC-MS/MS analysis can only speculate that HAP2 or 3 proteins are involved in regulating the kinase/phosphatase-involved cascades. The authors need to validate them to support their hypothesis. Does each HAP protein have a unique function or do they have the same function- forming a complex to regulate SnRK1?

Line 212: what is the accession number for SnRK2? I can't find it in the Data S5. It is difficult to look for it in the Data S5 without probably labeling it.

Fig. 5. Does the hap2 or hap3 mutant-infected maize show a reduction in starch accumulation similar to the hap1 mutant?

Line 376-378: I cannot agree with the authors' assertion that the HAP effectors function in a compensatory manner. How do the HAP effectors compensate for each other within the HAP complex? they can form a dimer? While forming the HAP complex is important for the induction of hypertrophy in maize leaves, it is not known whether HAP2 and HAP3 can also interact with SnRK1 or have different targets.

Reviewer #4

(Remarks to the Author)

In this study, Lee et al. showed that the *U. maydis* effectors Hap1-3 are involved in fungal virulence, by affecting the activity of SnRK1 in *Z. mays*. It led to alteration in cell cycle regulation and starch biosynthesis. The relationship between pathogen virulence and host metabolism emerges as an important topic in plant-pathogen interactions. Although their works are fascinating, this reviewer thought that the puzzle is not yet complete. They need to show more direct evidence to support their model.

Hap1 interacts with SnRK1. Phosphorylation of SnRK1 substrates decreases during the infection of hap1-CR, compared to SG200. However, these results do not directly support their model in which Hap1 activates SnRK1 during infection. How do Hap1-3 act for it? Although Hap effectors modify protein-protein interactions for Hap1, do they affect interactions about Hap1-SnRK1 or SnRK1-substrates? Do Hap effectors affect SnRK1 activity?

They claimed in this study that SnRK1 activity suppresses plant defense against *U. maydis*. Does this fungal virulence increase when SnRK1 is activated (for example, under dark conditions)? If so, can it be reduced by supplying sugars to suppress SnRK1?

Hap effectors bind each other, and they affect the PPI network in Fig 4. The PPI network is disconnected in Hap1-CR-Hap2/3, compared to Hap1-CR-Hap2. For example, how different are the PPI profiles between Hap1-2HA and Hap1-CR-Hap2? This information is required because the virulence of hap triple mutants is similar to that of hap1 mutants. What PPIs are the most important for Hap1 functions?

In Fig 5(a), red and blue indicate CR-hap1 and SG200, respectively. Is this correct? This reviewer thought that red and blue indicated up-regulated and down-regulated, respectively.

For example, in Fig6 b and c, "CR-Hap1" should be amended to "CR-hap1". These mistakes are found in the manuscript.

Version 1:

Reviewer comments:

Reviewer #1

(Remarks to the Author)

The authors have revised the manuscript very carefully and addressed all the concerns raised. The current version is acceptable for publication in Nature Communication.

Reviewer #2

(Remarks to the Author)

The revised manuscript is significantly improved and most of my previous comments had been suitably addressed during revision, I have a few minor points which can be addressed:

1. The data presented in the study, reveals that HAP1 contributes to starch biosynthesis. Please discuss if modulation in starch biosynthesis has been previously shown to be involved in hypertrophy.
2. Data related to phosphorylation status of SnRK in HAP1 and other effector mutants is not clear. Please clarify whether HIP1/Hip2 facilitates the physical interaction of HAP1 with SnRKs or whether their interaction alters the phosphorylation status of SnRK.

Reviewer #3

(Remarks to the Author)

The authors have included substantial new data in the revised manuscript, however, it remains insufficient to establish the biological relevance of the SnRK1–Hap1–Hip1–Hip2 complex in regulating starch accumulation and promoting HTT. The current study shows that Hap1 interacts with Hip1, Hip2, and the maize protein SnRK1; however, the functional significance of this complex formation is not convincingly demonstrated. The omics data offer only indirect support for the involvement of Hap and Hip proteins in host starch metabolism and do not show that Hap1, Hip1, and Hip2 inhibit SnRK1 activity by affecting its phosphorylation or that of downstream targets, nor that this suppression modulates immune responses. Additionally, the proposed redundancy between Hip1 and Hip2 is not well-supported by the data. At a minimum, the authors should complement the Δ hip1 mutant with Hip2 to determine whether the virulence phenotype can be rescued. Furthermore, while the authors suggest that Hip1 and Hip2 mediate or stabilize the SnRK1–Hap1 interaction through complex formation, no solid evidence is provided to support this mechanism.

- How do the authors explain the discrepancy in PP1 data between the luciferase and CoIP assays shown in Fig. 3?
- What is the virulence phenotype of Δ hip1 (00792)?
- If Hip1 and Hip2 have redundant functions, as proposed in lines 247–248, the minor reduction in starch accumulation observed in the Δ hip1 mutant could be due to compensation by Hip2. However, why was SnRK1 not pulled down in the Hap1-CR-hip1 or Hap1-CR-hip2 samples? If Hip1 and Hip2 do not have redundant roles, how do the authors account for the only minor starch reduction in Δ hip1?
- Line 308. Should be ZmSnRK1 α 1 or ZmSnRK1 α 3, right?
- It is not clear if the signals observed are full-length or truncated forms. Please provide immunoblots for colocalization experiments.
- In the colocalization assay, signals for Hap1 and SnRK1 α 2 were still observed even in the absence of Hip1 and Hip2. What are these detected signals representing? Moreover, the SnRK1 α 2 band was barely visible in the immunoblot. In contrast, Hap1 was strongly detected when it was co-expressed with GFP alone. The authors need to provide solid evidence to support the claim that the stability of Hap1 or SnRK1 α 2 is enhanced by Hip1 and Hip2.
- Do the authors have evidence supporting the translocation of Hap1, Hip1, or Hip2 from the fungus into plant cells?

Reviewer #4

(Remarks to the Author)

I agree that the revised manuscript is improved. However, the molecular mechanism by which Hap1 suppresses plant defense during infection remains unclear, which I consider essential for this study to address, even if only with a single piece of evidence. The authors have only shown that Hap1 interacts with ZmSnRK1, but this is not directly connected to the metabolic disorder observed during SG200 infection. Does Hap1 regulate SnRK1 activity positively or negatively? Have the authors attempted an in vitro kinase assay to test this? This should be a relatively straightforward experiment to perform.

While Fig. S12 provides important data suggesting that Hip1 and Hip2 stabilize Hap1 and ZmSnRK1, the quality of this figure is insufficient for publication. For example, there is not even a loading control presented, which makes it impossible to assess protein amounts and undermines the reliability of these results.

Fig. 6e is very confusing. The authors should explain this data more clearly in the manuscript. It is also unclear why they discuss proteins not shown in Fig. 6e. If these proteins are critical to their study, they should be included in the main figures.

Minor comments

1. The authors should carefully proofread the revised manuscript. Although they changed the names of the effectors to Hip1 and Hip2, I still found references to Hap2 and Hap3, especially in the figure legends.
2. The figure title or legend for Fig. 1 should clearly state that Hap1 corresponds to UMAG_02473. Without this information, readers will have difficulty following the manuscript. The same applies to Hip1 and Hip2; their corresponding gene identifiers should be indicated to improve clarity. In Fig. 5, why does the contribution of Hap1, Hip1, and Hip2 appear as a circle? Does this mean that Hip1 has a larger effect than the other two effectors?
3. In Fig. 5a, is “Hap1-2HA-CR-hip2” actually “Hap1-2HA-CR-hip1/2”?
4. In Fig. 5b, the names should be consistent with those in Fig. 5a.

5. In Fig. 6e, what do the asterisks indicate? Also, why are some Log2FC values shown in grey?

Version 2:

Reviewer comments:

Reviewer #1

(Remarks to the Author)

The manuscript has been revised effectively, and the latest version is now suitable for publication. The only minor concern I have is that the figures appear to be out of order in the revised file—I'm not certain whether I received the correct version.

Reviewer #2

(Remarks to the Author)

I have gone through the modified manuscript and with including of new data, the conclusion looks more convincing. From my side it is now suitable for publication.

Reviewer #3

(Remarks to the Author)

I thank the authors for taking my comments into account and for addressing most of my previous concerns. Below are a few minor points that can be considered:

Line 81-82: This statement is inaccurate. As reported in the cited reference, TaFROG is a wheat orphan protein that protects TaSnRK- α from degradation mediated by the *F. graminearum* effector Osp24.

Fig 3a. I concur that Hap1 translocates into host cells; however, the protein band observed in the nuclear fraction is larger than expected and may be nonspecific, particularly in the absence of a negative control. I therefore suggest rewording this statement.

Fig 6d-f: These co-IP figures should be combined and presented as a single figure.

Reviewer #4

(Remarks to the Author)

The revised manuscript has been improved overall. I have a few remaining comments below.

In Fig 1f, starch accumulation was reduced in CR-hap1–infected leaves compared to SG200-infected leaves. However, compared with mock-treated leaves, starch accumulation appears dispersed. This point should be explicitly described and discussed in the manuscript.

In line 283, the authors claim that Hip1/2 functions are not redundant with Hap1. This statement is confusing to me, because if their functions are indeed non-redundant, additive or synergistic effects would be expected. The authors should clarify the logic behind this interpretation.

In Fig3a, the nuclear Hap1 band appears to be shifted. This observation should be mentioned and interpreted in the manuscript.

In Fig6e, is ZmSnRK1a2 stabilized in the presence of Hip1? The authors should clarify this point in the text.

The plant material used for the assay of Fig7f is not clearly described. The authors must explicitly state which plants were used in this experiment.

Response to Reviewers

First, we would like to thank the four expert reviewers who provided us with in-depth, constructive feedback on our work. Their comments clearly revealed significant weaknesses in the paper.

Therefore, we have decided to streamline the paper and shortened weaker sections to clarify the core statements. We also conducted several new experiments and analyses to address the reviewers' questions, as far as was technically possible.

To provide a better overview, we have summarized the most important changes in bullet points for all reviewers. This is followed by a detailed, point-by-point response to each of the reviewers' comments.

Summary of major changes:

1. We substantially revised the Result and Discussion sections for better focus and flow:
 - i) The subsection "Characterization of Candidate Genes Associated with Hypertrophic Tumor Formation" was restructured to improve logical progression.
 - ii) The discovery of Hip1 and Hip2 now follows a revised storyline. It starts with Hap1, followed by the identification of its interactors, Hip1 and Hip2 (formerly Hap2 and Hap3).
 - iii) The earlier claim that Hap1/Hip1/Hip2 form a complex has been removed; Hip1 and Hip2 are Hap1-interacting proteins that enhance the Hap1-SnRK1 interaction.
 - iv) The previous statement that "Hap1 prevents SnRK1 inhibition by high levels of trehalose-6-phosphate (T6P)" has been removed. Instead, we propose that Hap1 acts as an SnRK1 inhibitor, as evidenced by the phenotypes (see Discussion, lines 407–415).
2. Disease test results for all HTT-related effectors. New data has been added (see **new Figure S1**).
3. Expanded Gene Ontology (GO) analysis of transcriptomic data (**new Figures S3-S5 and Supplementary Data 4-5 and 7**).
4. New starch accumulation phenotypes of the Hip2 mutant and the Hap1/Hip1/Hip2 triple mutant (**new Figure S9**).
5. Luciferase complementation assays were performed to demonstrate the specific interaction between Hip1/Hip2 and SnRK1 α 2 subunits (**new Figure S10bc**).
6. The interaction between Hip1/Hip2 and SnRK1 α 2 was further validated by co-immunoprecipitation experiments (**new Figure S10def**).
7. New co-localization experiments of SnRK1, Hap1, Hip1, and Hip2 were conducted to further support the interaction (**new Figure S11**).
8. A Western blot was performed to determine whether Hip1 and Hip2 influence the accumulation levels of Hap1 and SnRK1 α 2 proteins (see **new Figure S12**), which supports the idea that Hip1 and Hip2 can enhance the Hap1-SnRK1 interaction.
9. The model was revised according to the revisions (see **edited Figure 7**).

REVIEWER COMMENTS

Reviewer #1 (Remarks to the Author):

In this manuscript, the authors focus on three effectors, Hap1-3 (hypertrophy-associated proteins), which were identified as virulence factors promoting hypertrophic mesophyll tumor cells (HTT), with Hap1 identified as a key virulence factor. Hap proteins form protein complex, and Hap1 interacts with maize SnRK1, disrupting energy regulation. Transcriptomic, proteomics and phosphoproteomics show that Hap1 targets SnRK1, reprogramming host metabolism and energy for hypertrophy. The topic is very interesting. However, some of the conclusions are not well supported by the data, especially the relationship between energy metabolism and *Ustilago maydis* is mostly based on omic-data analysis, and lacks solid gene functional study.

The authors used 2xHA-complemented strains for IP and subsequent mass-spectrometry assay. The authors stated that they used infected maize leaves for protein extraction, which contain both the pathogen and plant cells. Based on the mass results, the author concluded that Hap proteins form complex in planta. Since this is a mixed tissue, the identified candidate interacting proteins cannot be simply concluded that they interact in planta, although it was validated that Hap protein can indeed interact with each other. Maybe Hap effectors only interact inside *U. maydis* and do not interact in plant cells. This need to be addressed. If Hap proteins form complex in planta, what is the meaning of Hap effector interacting with each other inside plant cells?

=> We thank the reviewer for raising this important point. It is correct that whole-leaf extracts contain both *U. maydis* and plant proteins, IP-MS alone cannot demonstrate that Hap effectors interact within plant cells rather than only during fungal growth. To address this, we have carried out in planta interaction assays in *Nicotiana benthamiana*: Split-Luciferase Complementation (LUC), Co-Immunoprecipitation (Co-IP) and Subcellular Co-Localization, and validated the interactions between effectors.

While our assays strongly support that the effectors interact in plant cells, we acknowledge they do not exclude the possibility that the primary interaction occurs in *U. maydis* before or during secretion. Live imaging constraints: technical limitations preclude real-time visualization of these interactions in maize tissue during infection. We have clarified in the revised manuscript that “in planta” refers to transient expression assays in *N. benthamiana* (Figure 3 and Figure 4: we added “following co-expression assays in *N. benthamiana*”), and we have noted the remaining uncertainty about the precise timing and location of interactions during the natural infection process (*Discussion, last paragraph, line 443-446*).

Following up the above questions, does Hap1 protein interact with Hap1 to form homodimers?

=> We co-expressed N-terminal and C-terminal luciferase fusions of Hap1 (nLuc-Hap1 and cLuc-Hap1) in *N. benthamiana*. Under the same conditions that effectively detect Hap1–Hip1 (formerly named as Hap2), Hap1–Hip2 (formerly named as Hap3) and Hip1–Hip2 interactions, we observed no significant luminescence signal from the Hap1–Hap1 pair. This indicates that there is no interaction between Hap1 and Hap1 to form homodimers.

The authors claim that Hap1 is the dominant effector based on Figure S3, however, based on Figure 1b, all three Hap genes are required for full virulence. Does the authors compare the virulence of Hap2 and Hap3 single mutant with the Hap1/2/3 triple mutant?

=> Indeed, all three effectors are required for virulence. Because both Hip1 and Hip2 (formerly named as Hap2 and Hap3) co-immunoprecipitated with Hap1, and share tumor-specific expression; Hip1 and Hip2 paralogous sequences, and a common promoter suggesting coordinated regulation, we hypothesized that Hap1 may functionally interact with the other two effectors (now referred as Hip1 and Hip2 in the manuscript), one aspect could be virulence, so we tested the triple mutant of Hap1, Hip1 and Hip2 to see if enhance virulence. The virulence phenotype of the Hip1 and Hip2 single mutants was not significantly different from that of the Hap1/Hip1/Hip2 triple mutant. We have therefore retracted our statement that Hap1 is a dominant effector in the manuscript.

The authors states that Hap1 interact with SnRK1 α 2 and SnRK1 α 3 (line 171-173), but no data was shown for Hap1 and SnRK1 α 3 (Fig .3a,b). Based on Fig 3c, Hap1 also interact with SnRK1 α 1? Why the interaction is inconsistent by different interaction systems?

=> Thank you for pointing this out. It was our mistake. A very weak (or even nonexistent) luminescent signal was detected after co-expressing Hap1 and SnRK1 α 3, which is not possible to be visualized in the figure. In the revised manuscript, we have corrected the statement on lines 171–173 and removed (it is now on lines 190-192). The LUC system requires sustained proximity of the N- and C-terminal luciferase fragments to reconstitute enzyme activity and emit light. While Hap1–SnRK1 α 2 produces a robust luminescent signal, the Hap1–SnRK1 α 3 interaction is below the LUC detection threshold, consistent with the CO-IP result (very weak bands of SnRK1 α 1 and 1 α 3 were detected after Co-IP, Figure 3d). Co-IP can capture both strong and weaker or transient interactions, provided the binding occurs during the extraction and pull-down steps.

In Figure S4, SnRK1 α 4 is also there, why the authors did not test the interaction between Hap1 and SnRK1 α 4?

=> We appreciate the reviewer's attention to Figure S4. Unfortunately, despite repeated efforts, we were unable to clone SnRK1 α 4 for our interaction assays. From our IP/MS results (Supplementary Data S8, Raw 53), SnRK1 α 2–4 are in the same protein group that interacts with HAP1. SnRK1 α 2, α 3, and SnRK1 α 4 share high sequence similarity, particularly in their kinase domains, which leads to overlapping peptide identification in mass spectrometry. In the revised manuscript, we focused on SnRK1 α 2, particular because the other Hip proteins appear to strengthen the interaction between Hap1 and SnRK1 α 2. SnRK1 α 2 not only has the strongest interaction with Hap1 but also represents the most interesting one among SnRK1 α subunits. Further experiments could explore whether SnRK1 α 4 also interacts with HAP1 directly, or if its detection was due to sequence overlap or co-complex formation.

Does Hap2 and Hap3 also interact with SnRK1?

=> Yes. We have confirmed the interaction between Hip1 or Hip2 (formerly named as Hap2/ 3) with SnRK1 α 2 by split-luciferase assay and CO-IP. The results are now presented on lines 293-312 and in the **new Figure S10**.

Protein interaction confirmation using only LCI and Co-IP is not enough. The authors need to do BiFC / Pull-down or yeast two hybrid assay to confirm the interactions.

=> We appreciate the reviewer's suggestion to include additional methods such as BiFC, pull-down, or yeast two-hybrid (Y2H) assays to further validate the Hap1–SnRK1 α interactions. The interaction was initially identified through IP-MS, and subsequently confirmed using two independent and complementary approaches: LCI and Co-IP, which are well-established and widely accepted methods for validating direct protein–protein interactions in planta.

In addition to these assays, we have now included microscopy-based co-localization data to further support the physical association of these proteins. These results are presented in the **newly added Figure S11**, which shows that Hap1, Hip1, Hip2, and SnRK1 α exhibit overlapping subcellular localization patterns. We fully agree that using other methods enhances the robustness of interaction data. But given the evidence from IP-MS, LCI, Co-IP, and co-localization, we believe our current data sufficiently support the interaction without redundancy. We remain committed to using the most appropriate and informative techniques to support our conclusions.

The authors did a lot of omic analysis, such as RNA-seq, proteomic and phosphoproteomic, and try to make solid conclusion from the omic data analysis. Although the data analysis seems support the conclusion perfectly, however, this is simply too ambitious and need point-to-point validations. The authors need to focus on key genes and make gene mutant of that gene(s) and functionally validate it (the author need to test disease phenotypes of target gene mutants). This is a major issue of this manuscript.

=> We acknowledge the importance of point-to-point validation the roles of key genes identified by omics. In our study, we focused on Hap1 due to its potential involvement in promoting hypertrophy in maize. To access this, we examined whether Hap1 influences starch accumulation and endoreduplication, both hallmark features of hypertrophy as previously described by Matei et al. (2018). Our transcriptomic data suggested that Hap1 can notably repress defense-related genes and promote starch metabolism. That's why we investigated the involvement of SnRK1. It would be great to validate via a maize SnRK1 mutant, but this process is time-consuming, takes over two years and therefore cannot be part of the present study. We have revised the manuscript to temper our conclusions, acknowledging the need for further studies (line 447-449).

Reviewer #2 (Remarks to the Author):

The manuscript describes how *Ustilago* disrupts the carbohydrate signaling network to induce hypertrophy in host cells. They identified three hap effector molecules and demonstrate that Hap1 potentially targets Snf1-related kinase 1 (SnRK1) and disrupts T6P-SnRK antagonistic relationship to regulate cell cycle and starch biosynthesis in the host.

The manuscript is mostly descriptive and based on proteomics and transcriptomics data. Although it demonstrates the interaction between the effector and the target host proteins, but the relevance of the interaction with loss and gain of functions studies is missing. Following are some of the major concerns about the manuscript.

1. The English language is not clear, and difficult to understand. The manuscript should be simplified by focusing on the physiological relevance of Hap1 interaction with SnRK1 and how this affects T6P function to modulate starch metabolism. Also, how hypertrophy is modulated during Hap1-SnRK1 interaction should be clearly demonstrated.

=> We have carefully revised the manuscript and, in response to the suggestion, we have also simplified the storyline. In short, our study aims to understand how Hap1 promotes hypertrophy in maize. We found that loss of Hap1 leads to a reduction in starch accumulation and downregulation of starch metabolic genes, suggesting a link between Hap1 and host carbohydrate metabolism. To identify possible targets, we performed IP/MS and identified SnRK1 subunits among the top Hap1 interactors. SnRK1 is a central regulator of energy balance and carbohydrate signaling in plants. Phosphoproteomic analysis revealed that phosphorylation of SnRK1 subunits and canonical SnRK1 consensus motifs is reduced in hap1 mutant infections compared to SG200. This suggests that Hap1 is required for full SnRK1 signaling output, potentially by promoting SnRK1 complex formation or substrate accessibility.

However, we acknowledge that our current data do not directly measure SnRK1 kinase activity or T6P levels. While T6P is a known inhibitor of SnRK1, we did not assess T6P function in this study. Our focus was on defining the host processes modulated by Hap1 and identifying SnRK1 as a potential integrator of this regulation. Future work will address whether the Hap1-SnRK1 interaction affects T6P signaling or SnRK1 enzymatic activity more directly. We have revised the Results and Discussion sections to emphasize these points and to clarify how changes in SnRK1 signaling correlate with altered starch metabolism and reduced hypertrophy in hap1 mutants.

2. The rationale of selecting Hap1-3 as a candidate effector paralogs is not clear. The table 1 enlists of some of the effectors, but not clear on what basis these genes were selected from the list reported in Rizzi et al and Zuo et al. The data related to knockout and complementation, with respect to the pathogenesis of *Ustilago* for each of the characterized effector genes, should be provided.

=> We thank the reviewer for pointing out the selecting threshold. The ten candidates were initially selected from a set of HTT-specific genes identified in previous study (Matei et al., 2018), these genes are cell type specific genes, only expressed in hypertrophic tumor (HTT) cells but not in hyperplastic tumor (HPT) cells. We have revised the manuscript and put more details in the introduction. Among the ten HTT-specific effectors, we prioritized Hap1 for detailed study because its knockout caused the most remarkable reduction in virulence. Subsequently, using IP-MS, we identified that Hip1 and Hip2 (formerly named as Hap2 and Hap3 in the manuscript)

interact with Hap1, prompting us to examine their roles as well. Hip1 was also one of the original HTT-specific candidates, while Hip2 shares a promoter region with Hip1, suggesting potential co-regulation. Following the reviewer's suggestion, we have now included additional disease test result in **Figure S1**.

3. The biochemical functions of Hap1-3 is not described in the manuscript. Also, the phylogenetic analysis and evolutionary conservation of these genes should be briefly described, for better understanding.

=> We appreciate the reviewer's suggestion to explore the biochemical function of Hap1 in more depth. We fully agree that elucidating the precise molecular activity of effectors is an important goal in understanding host-pathogen interactions. However, the primary focus of this study is to identify and validate the host targets of Hap1, particularly the SnRK1 α subunits, and to demonstrate the relevance of these interactions during *U. maydis* infection. Regarding phylogenetic analysis, Hap1 does not share significant similarity with Hip1 and Hip2 (formerly named Hap2 and Hap3), so we did not include a phylogenetic analysis. We have, however, clarified in the manuscript that Hip2 is a paralog of Hip1, and we have added the sequence similarity between these two effectors in the manuscript (line 208-209).

4. The physiological relevance of Hap effector's interaction with each other in planta, is not clear. Do they interact in a complex with the target host proteins? It would be more relevant to focus on unraveling the common interacting partners of Hap1 paralogs in the host.

=> In the revised manuscript, we have added experimental data showing that Hip1 and Hip2 (formerly Hap2 and Hap3) also interact with SnRK1 α 2 (Line 293-309, **new figure S10**). Moreover, all three effectors co-localize with SnRK1 α 2 in *N. benthamiana*, and Hip1 and Hip2 enhance the interaction between Hap1 and SnRK1 α 2 when co-expressed (Line 310-319, **new figure S11-12**). Despite these findings, we cannot conclude that these effectors form a multi-effector-host protein complex in planta. In light of this, we have revised our manuscript to clarify our interpretation: rather than stating that the three effectors form a complex, we now describe Hip1 and Hip2 as Hap1-interacting proteins that enhance Hap1-SnRK1 interactions. Whether these interactions occur simultaneously with SnRK1 or involve dynamic or sequential associations remains unknown. The formation of a stable multi-effector-host complex is an open question that will require further investigation.

We appreciate the reviewer's suggestion to explore potential common interactors of Hap1 paralogs. We performed a BLAST search of the *U. maydis* genome and identified one additional candidate, UMAG_02473, which shares ~27% sequence identity with Hap1. However, this gene was not identified among the hypertrophic tumor (HTT)-specific candidate effectors in the dataset published by Matei et al. (2018), and therefore was not prioritized for analysis in the current study.

5. Have you tested whether Hap2 or Hap3 interact with the SnRK1 protein in planta. In lines, 199-200 on page 8, it is mentioned that knockout mutants of Hap1 interact with the SnRK substrate, this needs to be properly described.

=> Thank you for the question, we have added experiments to test the interaction via a split-luciferase complementation assay and CO-IP between Hip1 and Hip2 (formerly named as Hap2

and Hap3) with SnRK1 (Line 293-309, and **new figure S10**). We have properly described the mutants on lines 278-280.

6. The role of Hap1 in modulating starch reprogramming and endoreduplication is mostly obtained from the transcriptome analysis. The direct involvement of Hap1 in regulating these processes is not clear. It is possible the observed changes may be due to the pleiotropic effect of SnRK1 modulation.

=> We concede that the original presentation of the data may have caused confusion by placing transcriptomic and phenotypic analyses in the same paragraph. To clarify, the observed changes in starch accumulation and endoreduplication are supported by both transcriptomic and experimental data. Specifically, starch accumulation was assessed using Lugol staining, and endoreduplication was evaluated by staining maize leaf sections with propidium iodide and measuring nuclear size in mesophyll cells (line 98-120, Figure 1).

7. It will be helpful if the authors test the direct involvement of Hap1 protein in regulating the cellular processes in the host. In this regard, creating a modified Hap1 wherein the active site is mutated and testing the effect of the WT and variants on the SnRK1 activity and T6P functions should be helpful.

=> Thank you for the suggestion. Effector proteins with structural functions typically lack an enzymatic active site, making it challenging to pinpoint functional domains. But yes, a mutant that loses SnRK1 binding should also not complement virulence. As a potential approach, we considered identifying conserved residues through homology with *Sporisorium* orthologs that might be essential for interaction with SnRK1. The idea was to test *Sporisorium* ortholog for complementation and then make mutant as negative experiment. Unfortunately, these efforts were technically unsuccessful despite multiple attempts. We recognize the value of this experiment and consider it a high-priority goal for future studies.

8. Whether Hap1 interferes with the biochemical function of SnRK1 or T6P ? The pathological development of *Ustilago* on the SnRK1 and T6S mutants should be tested.

=> It would be valuable to explore whether Hap1 interferes directly with biochemical functions of SnRK1 or T6P, however, this is not essential. Our current data suggest that Hap1 is more likely to affect SnRK1 signaling via phosphorylation. While it would be beneficial to test if SnRK1 can interact with T6S, we have decided to better focus our paper and consequently we did not further address T6S due to insufficient evidence.

Reviewer #3 (Remarks to the Author):

The authors demonstrated that Hap1, Hap2, and Hap3 form a protein complex, with only Hap1 shown to interact with maize ZmSnRK1 α , its primary target. The triple hap deletion mutant displayed a virulence phenotype similar to the single hap mutants, suggesting that all three HAP proteins are equally important in the effector complex formation. However, the biological relevance of the complex formation and its interaction with ZmSnRK1 α has not been investigated in this study.

Additionally, the authors stated that Hap1 prevents SnRK1 inhibition by high levels of trehalose-6-phosphate (T6P), but this aspect was also not explored in this work.

The impact of the HAP complex on the regulation of SnRK1 activity remains unclear, despite the authors' efforts to investigate it using proteomics, transcriptomics, and phosphoproteomics. Based on the GO and PPI results, there are both similarities and differences in the GO terms identified for the individual mutants, which may suggest functional similarities and differences among the HAP proteins. Unfortunately, the manuscript does not investigate whether Hap1 has a unique function distinct from Hap2 and Hap3. If all three HAP proteins share the same function—forming a complex to regulate SnRK1—I would expect to see similar outcomes in their phosphoproteomics studies and the starch accumulation levels in the single hap deletion mutants.

ZmSnRK1 is reported to play a defensive role; however, its mechanism of defending against pathogens by mobilizing energy remains unclear. While the authors demonstrated reduced starch accumulation in the hap1 mutant, no direct connection between Hap1 and SnRK1 has been established. The authors should address this by investigating starch accumulation patterns in SnRK1 deletion or overexpression transgenic lines.

- What is the significance of forming the Hap effector complex in interacting with ZmSnRK1 α or regulating the ZmSnRK1 α activity?

=> We agree, the complex formation has not been investigated in the study. We therefore have revised the manuscript, which now contains less interpretation and instead focuses on the direct experimental evidence. Consequently, rather than stating that the three effectors form a complex, we now describe Hip1 and Hip2 (formely named as Hap2 and Hap3) as Hap1-interacting proteins that enhance Hap1–SnRK1 interactions. Whether these effectors form a stable multi-effector–host complex or act independently to modulate SnRK1 α activity remains an open question and a promising direction for future research.

- Can the interaction of HAP1 affect the phosphorylation of SnRK1?

=> Our phosphoproteomic data suggest that the interaction of Hap1 influences the phosphorylation status of SnRK1 α 1, β 1 and β 2, and γ , although the effect does not appear to be through direct modification of the canonical activation site (the T-loop) of SnRK1. We cannot state whether Hap1 directly affects phosphorylation of SnRK1. Hap1 likely affects the phosphorylation of SnRK1 or components of the SnRK1 complex, although not in the conventional way one might expect (via direct phosphorylation of the T-loop).

- Can Hap2 or Hap3 interact with ZmSnRK1 α ?

=> Yes. We have confirmed the interaction between Hip1 or Hip2 (formerly named as Hap2/ 3) with SnRK1 α 2 by split-luciferase assay and CO-IP (Line 294-313, **new figure S10-11**).

- There is no evidence provided to demonstrate that Hap1 prevents SnRK1 inhibition by high levels of trehalose-6-phosphate (T6P)

=> We acknowledge that our study does not directly assess SnRK1 kinase activity or trehalose-6-phosphate (T6P) levels. Although T6P is a known inhibitor of SnRK1, we did not investigate its role in this work. Our primary focus was to define the host processes modulated by Hap1 and to identify SnRK1 as a potential integrator of these responses. To avoid confusion, we have removed previous references to T6P signaling and revised the Results and Discussion sections accordingly, as we realized that this was just fetching too far. We now emphasize the correlation between altered SnRK1 signaling and changes in starch metabolism and hypertrophy in hap1 mutants. Investigating whether the Hap1–SnRK1 interaction directly affects T6P signaling will be an important focus of future studies.

- It is known that ZmSnRK1 can form a complex, does the interaction with HAP1 disrupt the ZmSnRK1 complex or does HAP1 simply prevent ZmSnRK1 inhibition by T6P?

=> Honestly, we currently have no solid evidence showing whether the Hap1-SnRK1 interaction disrupts the ZmSnRK1 complex or if Hap1 prevents the inhibition of ZmSnRK1 by T6P. According to the IP/MS data, SnRK1 α 2-4 are in the same protein group that interacts with HAP1. SnRK1 α 1, however, is in a different group (see Supplementary Data S8; Raw53 contains the SnRK1 α 2-4 protein group). This pattern suggests that HAP1 may engage SnRK1 in more than one assembly state. SnRK1 α 1, β 1 and β 2, and γ were highly phosphorylated in S vs. M, but not in CR-H vs. M, indicating Hap1 is required for specific phosphorylation events on SnRK1 subunits, possibly Hap1 is required for SnRK1 complex formation.

Regarding the question about T6P, as you suggested above, no evidence was provided to demonstrate that Hap1 prevents SnRK1 inhibition by high levels of T6P. However, based on the phenotypes, our current data suggest that Hap1 inhibits SnRK1. Discussed in lines 409-414.

- ZmSnRK1 localizes to both nucleus and cytoplasm. Where does the interaction with HAP1 take place? This information is crucial for elucidating the mechanism by which the Hap complex regulates HTT formation for fungal virulence, but it is missing.

=> Thank you for the constructive suggestion, we have checked the co-localization of SnRK1 and the three effectors and found that they co-localize in both cytoplasm and the nucleus (in **new figure S12**).

Other comments:

Line 154-155: The three effectors form a complex, which is important for their virulence

function. The similar virulence phenotype in the triple and single mutants did not support that Hap1 has a dominant role.

=> Thank you for this comment - we agree. We therefore revised our statement, line: 244-246. We describe that: for the virulence, Hip1 and Hip2 (formerly Hap2 and Hap3) may act either redundantly with or downstream of Hap1 for virulence, rather than in parallel with Hap1.

Line 165: Please provide the accession numbers for ZmSnRK1 α subunits in the main text.

=> The accession numbers of ZmSnRK1 α subunits are now in Line 183-184.

Line 170-171: The split-luciferase complementation assay for the interaction of Hap1 and ZmSnRK1 α 3 is missing.

=> Thank you for pointing this out. It was a mistake in our manuscript. A very weak (or even nonexistent) luminescent signal was detected after co-expressing Hap1 and SnRK1 α 3, which is not possible to be visualized in the figure. In the revised manuscript, we have corrected the statement on lines 171–173 and removed (it is now on lines 190-192).

Fig 3c: Please show that the ZmSnRK1 α 1-3 proteins are not non-specifically bound to the Myc-beads.

=> In the Co-IP experiment, the specific interaction between SnRK-4myc and Hap1-6HA was demonstrated by reciprocal detection using both anti-Myc and anti-HA antibodies, which strongly supports specific binding. Additionally, Myc-beads are designed to specifically capture Myc-tagged proteins, and non-specific binding of ZmSnRK1 α 1-3 proteins to the beads alone is expected to be minimal, as consistently reported in the literature and confirmed by our negative controls where no HA-tagged protein was present. Therefore, we believe that additional demonstration of non-specific binding to Myc-beads is not necessary at this point.

Line 181-215 and Fig. 4: I don't understand how this analysis can support that the effector complex is important for orchestrating a regulatory cascade and mediating interactions between Hap1 and ZmSnRK1 α , since they have different GO terms. The data retrieved from the LC-MS/MS analysis can only speculate that HAP2 or 3 proteins are involved in regulating the kinase/phosphatase-involved cascades. The authors need to validate them to support their hypothesis. Does each HAP protein have a unique function or do they have the same function- forming a complex to regulate SnRK1?

=> We apologize for using confusing/misleading terminology with regard to a “protein complex”, which we indeed did not demonstrate. To avoid overinterpretation, we have revised the manuscript in this aspect. We now refer to Hip1 and Hip2 (formerly named Hap2 and Hap3) as Hap1-interacting proteins, without making conclusions about complex formation or shared regulatory cascades. About the functions, based on our current data, we cannot determine whether each effector protein (Hap1, Hip1, and Hip2) has a distinct function or whether they function together as a complex to regulate SnRK1. However, several observations provide initial insight: we have

tested the virulence of Hip1 and Hip2 and evaluated whether Hip1 promotes starch accumulation. Both effectors contribute to virulence, but the phenotypes were less pronounced than that of Hap1. The virulence phenotype of the triple knockout mutant (hap1/hip1/hip2) is comparable to that of the hap1 single mutant. Unlike Hap1, Hip1 had only a minor effect on starch accumulation. Together, this suggests that Hip1 and Hip2 may act either downstream of or redundantly with Hap1 during infection, rather than having independent or parallel roles.

Line 212: what is the accession number for SnRK2? I can't find it in the Data S5. It is difficult to look for it in the Data S5 without probably labeling it.

=> We have added the accession numbers (1.291): Zm00001eb434400, Zm00001eb392580.

Fig. 5. Does the hap2 or hap3 mutant-infected maize show a reduction in starch accumulation similar to the hap1 mutant?

=> Since Hip1 (formerly named Hap2) is HTT-specific and interacts more strongly with SnRK1 α 2, we tested the starch accumulation of the Hip1 mutant and the Hap1/Hip1/2 triple mutant. CR-sts2 was used as a positive control. CR-hap1 and CR-hap1/hip1/hip2 exhibited significantly reduced starch accumulation compared to SG200 in mesophyll, while CR-hap2 showed only a minor reduction. This part of the results have been added in the manuscript (line 246-254, **new Figure S9**).

Line 376-378: I cannot agree with the authors' assertion that the HAP effectors function in a compensatory manner. How do the HAP effectors compensate for each other within the HAP complex? they can form a dimer? While forming the HAP complex is important for the induction of hypertrophy in maize leaves, it is not known whether HAP2 and HAP3 can also interact with SnRK1 or have different targets.

=> Indeed, the term "compensatory" may not fully capture the relationship between Hap1, Hip1, and Hip2. Our data do not demonstrate that Hip1 and Hip2 can fully or partially replace Hap1's function in its absence. While Hip1 and Hip2 share some functions with Hap1, such as SnRK1 interaction (as confirmed by our additional experiments), they are insufficient to restore the same level of virulence as Hap1. The virulence level of the triple mutant of hap1, hip1, and hip2 was similar to that of the hap1 single mutant. We have therefore revised our statement.

In response to the inquiry regarding SnRK1 interactions, we conducted supplementary experiments demonstrating that Hip1 and Hip2 (previously designated as Hap2 and Hap3) interact with SnRK1 α 2 as well. As stated in our revised manuscript, Hip1 and Hip2 can strengthen the bond between Hap1 and SnRK1 α 2, which could potentially impact processes like starch metabolism.

Reviewer #4 (Remarks to the Author):

In this study, Lee et al. showed that the *U. maydis* effectors Hap1-3 are involved in fungal virulence, by affecting the activity of SnRK1 in *Z. mays*. It led to alteration in cell cycle regulation and starch biosynthesis. The relationship between pathogen virulence and host metabolism emerges as an important topic in plant-pathogen interactions. Although their works are fascinating, this reviewer thought that the puzzle is not yet complete. They need to show more direct evidence to support their model

Hap1 interacts with SnRK1. Phosphorylation of SnRK1 substrates decreases during the infection of hap1-CR, compared to SG200. However, these results do not directly support their model in which Hap1 activates SnRK1 during infection. How do Hap1-3 act for it? Although Hap effectors modify protein-protein interactions for Hap1, do they affect interactions about Hap1-SnRK1 or SnRK1-substrates? Do Hap effectors affect SnRK1 activity?

=> We agree that the phosphorylation data does not directly indicate that Hap1 activates SnRK1 during infection. We understand that the proposed model in the manuscript may have been somewhat confusing, so we have revised the relevant statements to clarify.

We acknowledge the limitation that we have not tested the activity of SnRK1 directly. According to the phosphorylation data, we actually do not claim that SnRK1 is activated by Hap1 during infection. Currently, it is unclear if SnRK1 is activated, nor can one conclude that Hap1 activates SnRK1 via direct phosphorylation. Instead, we propose that Hap1 is necessary for complete SnRK1 signaling, possibly by promoting SnRK1 complex assembly or increasing substrate accessibility. Furthermore, based on new experimental data, we have revised the manuscript to show that Hip1, Hip2, and Hip3 (formerly named Hap2, Hap3, and Hap4, respectively) enhance the interaction between Hap1 and SnRK1. However, there is no direct evidence showing whether Hip1 or Hip2 influence SnRK1's interaction with its substrates or activity.

They claimed in this study that SnRK1 activity suppresses plant defense against *U. maydis*. Does this fungal virulence increase when SnRK1 is activated (for example, under dark conditions)? If so, can it be reduced by supplying sugars to suppress SnRK1?

=> This is a very interesting perspective. Currently, no studies have been published about the virulence of *U. maydis* under dark conditions. SnRK1 is known to be activated under energy-deprivation conditions, such as darkness or sugar starvation. This activation generally promotes catabolic processes while suppressing biosynthetic and immune responses. Although the idea of testing whether supplying sugar under energy-depleted conditions affects virulence is mechanistically sound, experimental validation is lacking. This would be an interesting topic for a future study.

Hap effectors bind each other, and they affect the PPI network in Fig 4. The PPI network is disconnected in Hap1-CR-Hap2/3, compared to Hap1-CR-Hap2. For example, how different are the PPI profiles between Hap1-2HA and Hap1-CR-Hap2? This information is required because the virulence of hap triple mutants is similar to that of hap1 mutants. What PPIs are the most important for Hap1 functions?

=> We thank the reviewer for highlighting the importance of comparing the Hap1-2HA and Hap1-CR-Hap2 PPI profiles, especially given that the virulence of hap1/hip1/hip2 triple mutant is similar to that of hap1 single mutant. We agree that a detailed, side-by-side comparison could shed light on which specific interactions are critical for Hap1's function. However, the central aim of this study was to understand how *U. maydis* promotes hypertrophy in maize during infection, which led to the identification of Hap1 as a key virulence effector for this process. Our transcriptomic analyses revealed that hap1 mutant infections specifically show downregulation of genes involved in starch metabolic processes compared to wild-type SG200 infections. In parallel, IP-MS data from samples expressing Hap1 indicated an enrichment of the top proteins associated with protein phosphorylation and serine/threonine kinase activity. These findings led us to focus our investigation on SnRK1 subunits, which also appeared among the top 150 interactors in the Hap1 IP-MS dataset. Rather than exploring all possible interaction differences among the mutants, we chose to prioritize biological relevance by narrowing our focus to the Hap1-SnRK1 interaction. One remarkable finding from the PPIs of the mutants is that, in the presence of Hip1 and/or Hip2 (formerly named as Hap2 and Hap3), Hap1 no longer pulls down SnRK1 (line 255-269). The PPIs of Hip1/2 mutants provide a very important clue that leads us to hypothesize and confirm that Hip1/2 affect the interaction between Hap1 and SnRK1. Such a comparison would be informative and very much expand the scope of the current study; it would also shift attention away from our main focus. We therefore believe that the suggestion by the reviewer is a promising direction for future research.

In Fig 5(a), red and blue indicate CR-hap1 and SG200, respectively. Is this correct? This reviewer thought that red and blue indicated up-regulated and down-regulated, respectively.

=> Yes. Red (actually orange) is CR-hap1 and Blue is SG200.

For example, in Fig6 b and c, "CR-Hap1" should be amended to "CR-hap1". These mistakes are found in the manuscript.

=> Thank you for pointing this out. Has been corrected.

Summary of major changes

To provide a clearer overview, we present a summary of the most significant changes incorporated into this revised manuscript:

1. New experimental evidence providing direct link between Hap1 function and SnRK1 activity: SnRK1 activity assays were performed, showing that Hap1 reduces SnRK1-dependent phosphorylation of rACC *in-vivo* (**new Fig. 7f and g**)
2. New evidence for translocation of Hap1 into the host cells (**new Fig. 3a**)
3. New evidence showing that Hip1 and Hip2 increase the stability of Hap1 *in-vivo* (**new Fig. 6h and i**)
4. Generation of new recombinant *U. maydis* strains to test for functional redundancy of Hip1 and Hip2. Virulence phenotype of Hip1 (UMAG_00792) and complementation strain (CR-hip2/C(Hip1)) has been added (**new Figs. S8 and S9**)
5. New western blot detection was performed, confirming that the observed fluorescence of co-localization experiment originated from full-length proteins (**Fig. 6h**)
6. Revised conclusions:
 - a) the previous claim that Hip1 and Hip2 enhance Hap1-SnRK1 interaction has been revised to state that Hip1 and Hip2 increase the stability of Hap1.
 - b) The previous interpretation that Hip1 and Hip2 may act either redundantly with or downstream of Hap1 for virulence, rather than in parallel. It has been changed to: Hip1 and Hip2 do not function redundantly with Hap1 for virulence.

REVIEWER COMMENTS

Reviewer #1 (Remarks to the Author):

The authors have revised the manuscript very carefully and addressed all the concerns raised. The current version is acceptable for publication in Nature Communication.

=> We thank this reviewer for the positive feedback.

Reviewer #2 (Remarks to the Author):

The revised manuscript is significantly improved and most of my previous comments had been suitably addressed during revision, I have a few minor points which can be addressed:

=> We thank this reviewer for the positive feedback.

1. The data presented in the study, reveals that HAP1 contributes to starch biosynthesis. pl discuss if modulation in starch biosynthesis has been previously shown to be involved in hypertrophy.

=> we have added this to the discussion as indicated on lines 430-436.

2. Data related to phosphorylation status of SnRK in HAP1 and other effector mutants is not clear. Please clarify whether HIP1/Hip2 facilitates the physical interaction of HAP1 with SnRKs or whether their interaction alters the phosphorylation status of SnRK.

=> We followed these questions, which also relate to the points brought up by reviewers 3 and 4, with new experiments.

In the previous manuscript version, we hypothesized that Hip1/2 increase the binding of Hap1 with SnRK. However, (and we fully agree with this reviewer) we could not explain this observation on the mechanistic level. Our new data could better specify the role of Hip1 and Hip2 regarding Hap1 function. We found that the stability of Hap1 is increased in presence of Hip1/2 (see **new Fig 6h and i**). As shown in the newly added figure (Fig. 6h and i), when equal amounts of total protein were loaded, Hap1 accumulated higher when co-expressed with Hip1 and Hip2. Co-expression of Hip1 and Hip2 increase the detectable protein amounts of Hap1. The finding that presence of Hip1/2 stabilizes Hap1 explains why less interactors of Hap1 identified by IP-MS after knocking out Hip1 and/or Hip2 (Fig 5b, Data S8, S12-14), i.e., a lower Hap1 stability results in a lower number of identified interactors. Contrary, with Hap1 being stabilized in presence of Hip1 and Hip2, more interactors could be detected, including SnRK1

=> Whether the interaction between Hip1/2 and Hap1 alters the phosphorylation status of SnRK1 is an excellent question which we also addressed with new experiments. Our **new data (Fig. 7f and Fig. S13)** shows that Hap1 suppresses SnRK1-dependent phosphorylation of downstream substrate ACC, suggesting a potential effect on SnRK1 activity. It is likely that the suppression of SnRK1 activity by Hap1 is directly related to their physical interaction. Several cases have been reported showing that effectors inhibit kinase activity by directly binding the kinase, even without changing its phosphorylation state. Given that Hip1 and Hip2 enhance the stability of Hap1 and thereby promote Hap1-SnRK1 interaction, Hip1 and Hip2 likely help with suppression of SnRK1 activity. However, also with the current data, we cannot conclude if whether phosphorylation of SnRK1 itself is altered. Future experiments will be required to resolve this point.

Reviewer #3 (Remarks to the Author):

The authors have included substantial new data in the revised manuscript, however, it remains insufficient to establish the biological relevance of the SnRK1–Hap1–Hip1–Hip2 complex in regulating starch accumulation and promoting HTT. The current study shows that Hap1 interacts with Hip1, Hip2, and the maize protein SnRK1; however, the functional significance of this complex formation is not convincingly demonstrated. The omics data offer only indirect support for the involvement of Hap and Hip proteins in host starch metabolism and do not show that Hap1, Hip1, and Hip2 inhibit SnRK1 activity by affecting its phosphorylation or that of downstream targets, nor that this suppression modulates immune responses. Additionally, the proposed redundancy between Hip1 and Hip2 is not well-supported by the data.

At a minimum, the authors should complement the Δ hip1 mutant with Hip2 to determine whether the virulence phenotype can be rescued. Furthermore, while the authors suggest that Hip1 and Hip2 mediate or stabilize the SnRK1–Hap1 interaction through complex formation, no solid evidence is provided to support this mechanism.

=> We thank the reviewers for the detailed feedback and accept the criticism of missing, more direct evidence, particularly regarding a functional connection of Hap1 activity and SnRK1 downstream signaling.

We have addressed these points with new experimental data as follows:

1. We have now included a new experiment, which shows that SnRK1 downstream signaling is reduced in the presence of Hap1. By using an in-vivo activity SnRK1 assay adapted from Avidan et al. (2023) we found that phosphorylation of rACC1 is reduced by co-expression of Hap1 (see **new Fig. 7f**).

Whether this reduction of SnRK1 activity directly modulates immune response cannot be definitively concluded at this stage. However, based on published studies, SnRK1 activation is known to promote defense responses, as noted in lines 448-450. Consistent with this, our RNA-seq data show that in the absence of Hap1, defense responses are up-regulated. Together, these findings further strengthen our proposed model: in the presence of Hap1, SnRK1 signaling is suppressed, which in turn leads to a down-regulation of defense responses.

2. Regarding the 2nd comment that we proposed redundancy between Hip1 and Hip2, we would like to clarify that this was not our intention. As stated in lines 262-263, because the virulence level of triple mutant (Hap1, Hip1 and Hip2) was similar to Hap1 single mutant, we previously concluded: Hip1 and Hip2 may act either redundantly with or downstream of Hap1 for virulence, rather than in parallel. If Hip1 or Hip2 were redundant with Hap1, deleting both of them in addition to Hap1 would further reduce virulence, compared with the hap1 single mutant. Therefore, it is unlikely that Hip1 and Hip2 act redundantly with Hap1. Regarding the redundancy between Hip1 and Hip2, we realized that we did not intend to imply that Hip1 and Hip2 themselves are fully redundant. To directly address this concern, we have now also generated a strain in which Hip2 is genetically complemented the Hip1 knockout. However, this strain fails

to recover the Hip1-dependent virulence defect (**new Fig. S9**).

3. About how Hip1 and Hip2 stabilize the SnRK1–Hap1 interaction:

We accept that our previous version did not provide solid experimental evidence to support the formation of a stable complex between Hip1, Hip2, Hap1, and/or ZmSnRK1 α 2. As stated in lines 481-485, our current knowledge does not allow us to conclude that a functional protein complex exists inside host cells, and we highlighted that resolving the structure of these interactions is an important aim for future work. Regarding the specific point of how Hip1 and Hip2 influence or enhance the SnRK1–Hap1 interaction, **we now include additional experimental data (Fig. 6h and i)**. Further explanation is included in our response to the related comment below.

- How do the authors explain the discrepancy in PPI data between the luciferase and CoIP assays shown in Fig. 3?

=> Split-luciferase requires the two fusion proteins to come very close and in the correct orientation for luciferase activity to reconstitute. A weak or transient interaction may not produce enough signal in LUC, leading to a false negative result. Co-IP can detect weak or transient interactions because it pulls down the protein complex from lysates. Thus, even proteins that interact briefly or indirectly can potentially be detected, with this method which is more sensitive as compared to the Luciferase assay.

- What is the virulence phenotype of Δ hip1 (00792)?

=> We have included the virulence phenotype of CR-Hip1 in the **new Fig. S8**. The Δ hip1 mutant shows a significantly reduced virulence.

- If Hip1 and Hip2 have redundant functions, as proposed in lines 247–248, the minor reduction in starch accumulation observed in the Δ hip1 mutant could be due to compensation by Hip2. However, why was SnRK1 not pulled down in the Hap1-CR-hip1 or Hap1-CR-hip2 samples? If Hip1 and Hip2 do not have redundant roles, how do the authors account for the only minor starch reduction in Δ hip1?

=> As noted above, we did not intend to suggest Hip1 and Hip2 are functionally redundant. We have added new experimental data (**new Fig. S9**), which shows that Hip1 and Hip2 are indeed not redundant. Regarding the only minor starch reduction in the Hip1 knockout, this might be a consequence of a lower stability of Hap1 in the absence of Hip1. As shown in the newly added figure (**new Fig. 6h and i**), when equal amounts of total protein were loaded, Hap1 accumulated higher when co-expressed with Hip1 and Hip2. Hip1 and Hip2 increase the stability of Hap1, which likely contributes to its functional activity

- Line 308. Should be ZmSnRK1 α 1 or ZmSnRK1 α 3, right?

=> Exactly, thank you.

- It is not clear if the signals observed are full-length or truncated forms. Please provide immunoblots for colocalization experiments.

=> We have included **immunoblots in Fig. 6** showing this.

- In the colocalization assay, signals for Hap1 and SnRK1 α 2 were still observed even in the absence of Hip1 and Hip2. What are these detected signals representing?

=> We found the interaction between Hap1 and SnRK1 α 2 using multiple independent approaches: IP-MS during maize infection, split-luciferase and Co-IP after co-expression in *N. benthamiana*, and now we also observed we found that Hip1 and Hip2 can influence the stability of Hap1 (**new Fig. 6H and i**).

Our IP-MS results indicate that SnRK1 α 2 is not detected in the Hap1-CR-hip1 or Hap1-CR-hip2 inoculated maize plants, and the Western blot data shows that Hap1 accumulates more with the presence of Hip1 and Hip2 (Fig. 6h and i). This suggests that in the absence of Hip1 and Hip2, the stability of Hap1 is reduced, which could lead to a weaker Hap1-SnRK1 interaction that likely falls below the detection threshold by IP-MS in infected maize plants.

Overexpression increases local concentration and co-localization, so more protein interactions become detectable. This explains why we can still observe the co-localization signal of Hap1 and SnRK1 after co-expression in *N. benthamiana* even without Hip1 and Hip2.

Moreover, the SnRK1 α 2 band was barely visible in the immunoblot. In contrast, Hap1 was strongly detected when it was co-expressed with GFP alone. The authors need to provide solid evidence to support the claim that the stability of Hap1 or SnRK1 α 2 is enhanced by Hip1 and Hip2.

=> We thank the reviewer for this suggestion, which actually helped us to better specify the roles of Hip1 and Hip2 for Hap1 function. In the previous manuscript version, we hypothesized that Hip1/2 increase the binding of Hap1 with SnRK. However, we could not explain this observation on the mechanistic level. We found that the stability of Hap1 is increased in presence of Hip1/2 (**see new Fig 6h and i**). As shown in the newly added figure (Fig. 6h and i), when equal amounts of total protein were loaded, Hap1 accumulated higher when co-expressed with Hip1 and Hip2. Co-expression of Hip1 and Hip2 increase the detectable protein amounts of Hap1. The finding that presence of Hip1/2 stabilizes Hap1 explains why less interactors of Hap1 identified by IP-MS after knocking out Hip1 and/or Hip2 (Fig 5b, Data S8, S12-14), i.e. a lower Hap1 stability results in a lower number of identified interactors. Contrary, with Hap1 being stabilized in presence of Hip1 and Hip2, more interactors could be detected, including SnRK1

- Do the authors have evidence supporting the translocation of Hap1, Hip1, or Hip2 from the fungus into plant cells?

=>Thank you for the suggestion. We have included **new experimental data in Fig. 3a**. By isolating crude cytoplasmic and nuclear protein fractions from maize tissue after 3dpi inoculation with strain expressing Hap1-2HA, we were able to detect Hap1 in both cytoplasm and nucleus, indicating its translocation into host cells.

In addition, our maize IP-MS data was also generated from the samples at 3 dpi after inoculation. IP-MS shows that Hap1 pulls down Hip1, Hip2 and SnRK1, which means that Hip1, Hip2 are likewise translocated into the plant cell and reside in the same compartment with Hap1 during infection. This is also confirmed by the co-localization experiments shown in Fig. 6g and h.

Reviewer #4 (Remarks to the Author):

I agree that the revised manuscript is improved. However, the molecular mechanism by which Hap1 suppresses plant defense during infection remains unclear, which I consider essential for this study to address, even if only with a single piece of evidence.

=> We agree that understanding the molecular mechanism underlying Hap1-mediated suppression of plant defense is essential. Our analyses provide a consistent mechanistic framework supported by multiple lines of evidence.

First, the RNA-seq data revealed Hap1-dependent induction of starch biosynthesis and cell-cycle-related genes, and suppression of plant defense responses. To explore how Hap1 might coordinate these changes, we examined its host interactors and identified SnRK1 as a particularly interesting candidate.

SnRK1 is a central energy and stress regulator, when activated, typically represses anabolic metabolism (including starch biosynthesis) and activates catabolic and plant defense. This canonical SnRK1 response contrasts with the phenotype observed in Hap1-dependent HTT cells. Based on this contrast, we hypothesized that Hap1 might inhibit SnRK1 activity to suppress plant defense.

In the newly added experimental data, we now show that Hap1 suppresses SnRK1-dependent phosphorylation of downstream substrate ACC (new Fig. 7f,g), providing evidence that Hap1 suppresses SnRK1 kinase activity *in-vivo*. This suppression offers a mechanistic explanation for the reduced defense responses observed in the presence of Hap1.

Additional mechanisms may also contribute, but our data collectively establish SnRK1 inhibition as a biologically grounded and experimentally supported mechanism through which Hap1 modulates host defense during infection.

The authors have only shown that Hap1 interacts with ZmSnRK1, but this is not directly connected to the metabolic disorder observed during SG200 infection. Does Hap1 regulate SnRK1 activity positively or negatively? Have the authors attempted an in

vitro kinase assay to test this? This should be a relatively straightforward experiment to perform.

=> We thank the reviewer for this suggestion. We fully agree that directly assessing whether Hap1 regulates SnRK1 activity is important. To address this, we adapted SnRK1 kinase activity assay from Avidan et al., 2023. In the newly added experiment (Fig. 7f,g), Hap1 suppresses SnRK1-dependent phosphorylation of ACC, suggesting a potential inhibitory effect on SnRK1 activity.

While Fig. S12 provides important data suggesting that Hip1 and Hip2 stabilize Hap1 and ZmSnRK1, the quality of this figure is insufficient for publication. For example, there is not even a loading control presented, which makes it impossible to assess protein amounts and undermines the reliability of these results.

=> We fully agree. This is now shown in Fig. 6i in the revised manuscript.

Fig. 6e is very confusing. The authors should explain this data more clearly in the manuscript. It is also unclear why they discuss proteins not shown in Fig. 6e. If these proteins are critical to their study, they should be included in the main figures.

=> Thank you for the suggestion. We have now modified Fig. 6 accordingly.

Minor comments

1. The authors should carefully proofread the revised manuscript. Although they changed the names of the effectors to Hip1 and Hip2, I still found references to Hap2 and Hap3, especially in the figure legends.

=> We have corrected the names of effectors.

2. The figure title or legend for Fig. 1 should clearly state that Hap1 corresponds to UMAG_02473. Without this information, readers will have difficulty following the manuscript. The same applies to Hip1 and Hip2; their corresponding gene identifiers should be indicated to improve clarity. In Fig. 5, why does the contribution of Hap1, Hip1, and Hip2 appear as a circle? Does this mean that Hip1 has a larger effect than the other two effectors?

=> We have added the information. Regarding Fig. 5, the circular representation of Hap1, Hip1, and Hip2 contributions does not indicate any difference in effect; it is simply a way to illustrate the knockout content.

3. In Fig. 5a, is “Hap1-2HA-CR-hip2” actually “Hap1-2HA-CR-hip1/2”?

=> Thank you for pointing out. Has been corrected.

4. In Fig. 5b, the names should be consistent with those in Fig. 5a.

=> Thank you for pointing this out.

5. In Fig. 6e, what do the asterisks indicate? Also, why are some Log2FC values shown in grey?

=> Thank you for pointing this out. The legend is now included in Fig. 6e (now Fig. 7e).

REVIEWERS' COMMENTS

Reviewer #1 (Remarks to the Author):

The manuscript has been revised effectively, and the latest version is now suitable for publication. The only minor concern I have is that the figures appear to be out of order in the revised file—I'm not certain whether I received the correct version.

Reviewer #2 (Remarks to the Author):

I have gone through the modified manuscript and with including of new data, the conclusion looks more convincing. From my side it is now suitable for publication.

Reviewer #3 (Remarks to the Author):

I thank the authors for taking my comments into account and for addressing most of my previous concerns. Below are a few minor points that can be considered:

Line 81-82: This statement is inaccurate. As reported in the cited reference, TaFROG is a wheat orphan protein that protects TaSnRK-a from degradation mediated by the *F. graminearum* effector Osp24.

=> Thank you for pointing this out. Has been corrected.

Fig 3a. I concur that Hap1 translocates into host cells; however, the protein band observed in the nuclear fraction is larger than expected and may be nonspecific, particularly in the absence of a negative control. I therefore suggest rewording this statement.

=> Has been added

Fig 6d-f: These co-IP figures should be combined and presented as a single figure.

=> Figure has been modified as suggested.

Reviewer #4 (Remarks to the Author):

The revised manuscript has been improved overall. I have a few remaining comments below.

In Fig 1f, starch accumulation was reduced in CR-hap1-infected leaves compared to SG200-infected leaves. However, compared with mock-treated leaves, starch accumulation appears dispersed. This point should be explicitly described and discussed in the manuscript.

=> Has been added

In line 283, the authors claim that Hip1/2 functions are not redundant with Hap1. This statement is confusing to me, because if their functions are indeed non-redundant, additive or synergistic effects would be expected. The authors should clarify the logic behind this interpretation.

=> The respective statement has been revised in the results section.

In Fig3a, the nuclear Hap1 band appears to be shifted. This observation should be mentioned and interpreted in the manuscript.

=> Has been added.

In Fig6e, is ZmSnRK1a2 stabilized in the presence of Hip1? The authors should clarify this point in the text.

=> While the accumulation of SnRK1 appears increased in the presence of Hip1 in this experiment, the aim of this assay was to confirm the interaction between SnRK1 and Hips. SnRK1 accumulation was not systematically quantified, and variations in total protein loading were present in this experiment. We therefore refrain from drawing conclusions regarding a potential stabilizing effect of Hip1 on SnRK1.

The plant material used for the assay of Fig7f is not clearly described. The authors must explicitly state which plants were used in this experiment.

=> Information has been added in legend and methods section.